# Translational control of polyamine metabolism by CNBP is required for *Drosophila* locomotor function

**Sonia Coni[1†], Federica A Falconio[2,3†], Marta Marzullo[2,4†], Marzia Munafò[5], Benedetta Zuliani[2], Federica Mosti[2,6], Alessandro Fatica[2], Zaira Ianniello[2], Rosa Bordone[1], Alberto Macone[7], Enzo Agostinelli[8,9], Alessia Perna[10], Tanja Matkovic[11], Stephan Sigrist[11], Gabriella Silvestri[10,12], Gianluca Canettieri[1,9,13]\*, Laura Ciapponi[2]\***

[1]Department of Molecular Medicine, Sapienza University of Rome, Rome, Italy; [2]Department of Biology and Biotechnologies, Sapienza University of Rome, Rome, Italy; [3]Department of Life Sciences Imperial College London South Kensington campus, London, United Kingdom; [4]IBPM CNR c/o Department of Biology and Biotechnology, Sapienza University of Rome, Rome, Italy; [5]European Molecular Biology Laboratory (EMBL) Epigenetics & Neurobiology Unit, Campus Adriano Buzzati-Traverso, Monterotond, Italy; [6]Department of Neurobiology, Duke University Medical Center, Durham, United States; [7]Department of Biochemical Sciences, Sapienza University of Rome, Rome, Italy; [8]Department of Sensory Organs, Sapienza University of Rome, Policlinico Umberto I, Rome, Italy; [9]International Polyamines Foundation 'ETS-ONLUS', Rome, Italy; [10]Department of Neuroscience, Fondazione Policlinico Gemelli IRCCS, University Cattolica del S. Cuore, Roma, Italy; [11]Freie Universität Berlin, Institute for Biology and Genetics, Berlin, Germany; [12]Department of Scienze dell'Invecchiamento, Neurologiche, Ortopediche e della testa-Collo; UOC Neurologia, Fondazione Policlinico Universitario 'A. Gemelli' IRCCS, Rome, Italy; [13]Pasteur Institute, Fondazione Cenci-Bolognetti, Rome, Italy

**\*For correspondence:**
gianluca.canettieri@uniroma1.it (GC);
laura.ciapponi@uniroma1.it (LC)

[†]These authors contributed equally to this work

**Competing interests:** The authors declare that no competing interests exist.

**Abstract** Microsatellite expansions of CCTG repeats in the cellular nucleic acid-binding protein (*CNBP*) gene leads to accumulation of toxic RNA and have been associated with myotonic dystrophy type 2 (DM2). However, it is still unclear whether the dystrophic phenotype is also linked to CNBP decrease, a conserved CCHC-type zinc finger RNA-binding protein that regulates translation and is required for mammalian development. Here, we show that depletion of *Drosophila* CNBP in muscles causes ageing-dependent locomotor defects that are correlated with impaired polyamine metabolism. We demonstrate that the levels of ornithine decarboxylase (ODC) and polyamines are significantly reduced upon dCNBP depletion. Of note, we show a reduction of the CNBP-polyamine axis in muscles from DM2 patients. Mechanistically, we provide evidence that dCNBP controls polyamine metabolism through binding dOdc mRNA and regulating its translation. Remarkably, the locomotor defect of dCNBP-deficient flies is rescued by either polyamine supplementation or dOdc1 overexpression. We suggest that this dCNBP function is evolutionarily conserved in vertebrates with relevant implications for CNBP-related pathophysiological conditions.

## Introduction

Myotonic dystrophy (DM) is the most common inherited muscle dystrophy in adults and comprises two genetically distinct forms: DM type 1 (DM1, Steinert' disease, OMIM 160900), caused by an expansion of CTG repeats in the 3′ untranslated region of the DM protein kinase (*DMPK*) gene (*Brook et al., 1992*) and DM type 2 (DM2, OMIM 602668), due to the expansion of CCTG repeats in the first intron of the cellular nucleic acid-binding protein (*CNBP*) gene, also named *ZNF9* (zinc finger protein 9; *Liquori et al., 2001*). Both DM1 and DM2 display a multisystemic involvement of the skeletal muscle, heart, eye, brain, endocrine system, and smooth muscle. The similarities in the clinical features have led to the hypothesis of a common pathogenic mechanism, represented by toxic gain-of-function of RNAs transcribed from alleles containing expanded CUG or CCUG repeats. These RNAs are ubiquitously transcribed, folded into hairpin structures and accumulated in nuclear foci, affecting the function of RNA-binding proteins such as the muscleblind-like proteins (MBNL1-3) and CUG-binding protein 1 (CUG-BP1) that regulates alternative splicing (*Kanadia et al., 2006*; *Mohan et al., 2014*). Recently, the involvement of additional non-mutually exclusive mechanisms, such as bi-directional antisense transcription, alteration of microRNA expression, and non-ATG-mediated translation (RAN) have been demonstrated (*Cho et al., 2005*; *Juźwik et al., 2019*; *Moseley et al., 2006*; *Nguyen et al., 2019*; *Perbellini et al., 2011*). In particular, ectopic RAN translation has been reported in several degenerative diseases caused by microsatellite expansions such as SCA8 (spinocerebellar ataxia type 8), DM1, fragile X-associated tremor ataxia syndrome (FXTAS), C9ORF72 amyotrophic lateral sclerosis/frontotemporal dementia (ALS/FTD), Fuchs endothelial corneal dystrophy, SCA31 (spinocerebellar ataxia type 31), Huntington disease, and recently in DM2 (*Zu et al., 2011*; *Zu et al., 2013*; *Mori et al., 2013*; *Ash et al., 2013*; *Todd et al., 2013*; *Bañez-Coronel et al., 2015*; *Zu et al., 2017*; *Ishiguro et al., 2017*; *Soragni et al., 2018*; *Nguyen et al., 2019*).

The presence of a repeat expansion might also lead to loss-of-function of the protein encoded by the affected mRNA. Haploinsufficiency of *CNBP* gene, resulting from the nuclear sequestration and/or altered processing of expanded pre-mRNAs, has been proposed to play an important role in the pathogenesis of DM2. In mice, heterozygous deletion of one *CNBP* allele causes a phenotype reminiscent of DM2 myopathy that gets worse with age, while homozygous deletion causes muscle atrophy and severe locomotor dysfunction already in young mice (*Chen et al., 2007*; *Wei et al., 2018*). Studies on muscle tissues or myoblasts from DM2 patients provided controversial results: some studies found normal *CNBP m*RNA and protein levels in muscle tissues (*Raheem et al., 2010*), while recent findings documented decreased protein levels in muscle tissues (*Huichalaf et al., 2009*; *Raheem et al., 2010*; *Salisbury et al., 2009*; *Schneider-Gold and Timchenko, 2010*; *Wei et al., 2018*). The hypothesis that CNBP deficiency plays a key role in DM2 pathogenesis implies that perturbation of CNBP function contributes to this disease.

CNBP is a highly conserved ssDNA-binding protein of 19 kDa (*Calcaterra et al., 2010*) involved in the control of both transcription, by binding to ssDNA and unfolding G-quadruplex DNAs (G4-DNAs) in the nuclei, and translation, by binding to mRNA and unfolding G4-related structures in the cytosol (*Armas et al., 2008*; *Benhalevy et al., 2017*; *David et al., 2019*; *Huichalaf et al., 2009*; *Iadevaia et al., 2008*; *Leipheimer et al., 2018*). Additionally, CNBP promotes internal ribosome entry site (IRES)-dependent translation of the ornithine decarboxylase (ODC) mRNA working as IRES-transacting factor (ITAF; *Gerbasi and Link, 2007*; *Sammons et al., 2011*). In our previous studies, we found that CNBP promotes IRES-mediated translation of ODC and polyamine metabolism in neurons and that this mechanism is aberrantly activated in the medulloblastoma (*D'Amico et al., 2015*). Hence, these studies highlighted the ability of CNBP to control polyamine metabolism and illustrated the consequence of an aberrant function of this molecular regulatory mechanism in human disease.

Polyamines (putrescine, spermidine, and spermine) are ubiquitous positively charged aliphatic amines that control key aspects of cell biology, such as cell growth, cell death, replication, translation, differentiation, and autophagy (*Casero et al., 2018*; *Coni et al., 2019*; *Wallace, 2000*). Polyamine metabolism starts from the decarboxylation of ornithine into putrescine, then putrescine is converted into spermidine, which is in turn transformed into spermine (*Casero et al., 2018*; *Wallace et al., 2003*). Because of their critical role, the intracellular concentration of polyamines is tightly regulated. Conversion of ornithine into putrescine, catalyzed by ODC, an enzyme with an

evolutionarily conserved function, represents the rate limiting step (*Sammons et al., 2011*; *Wallace et al., 2003*). Indeed, the intracellular levels of ODC are promptly adjusted to the cellular needs, thanks to different mechanisms affecting its protein stability, transcription, and translation (*Pegg, 2006*). Alterations of polyamine content are found in different pathophysiological conditions such as cancer, degenerative disorders, and aging (*Casero et al., 2018*), although their specific role in muscle disorders has not been fully characterized yet.

Given the conservation of CNBP primary structure and function between *Drosophila melanogaster* and vertebrates/humans (*Antonucci et al., 2014*; *D'Amico et al., 2015*) in this work, we have investigated the effect of *CNBP* loss-of-function. We show that dCNBP depletion in muscles reduces fly viability and causes a robust locomotor defect. Furthermore, we demonstrate that dCNBP directly affects polyamine metabolism by regulating dOdc mRNA translation and, notably, that restoration of proper polyamine content rescues muscle function.

## Results

### *CNBP* is essential for fly locomotion

To explore the role of CNBP, we conducted RNAi-mediated knockdown experiments of *Drosophila dCNBP* gene. As previously shown, the expression of two copies of the RNAi construct (2XUAS*dCNBP*$^{RNAi}$) under the control of ubiquitous promoters resulted in embryonic or larval lethality (*Antonucci et al., 2014*). Thus, to address the in vivo function of *dCNBP*, we drove the expression of 2XUAS*dCNBP*$^{RNAi}$ using tissue-specific GAL4 drivers (*Table 1*). We did not observe any effect on viability or fly locomotion activity when *dCNBP* was silenced in neurons, motor neurons, or glia (*Table 1*). The efficacy of dCNBP knockdown with neuronal drivers was extremely efficient, as documented by dCNBP immunoblots from larval and adult brain lysates (*Figure 1—figure supplement*

**Table 1.** Effects of dCNBP silencing using different tissue-specific GAL4 drivers.

| Driver line | Expression pattern | 2X*dCNBP*$^{RNAi}$ at 29°C | 2X*dCNBP*$^{RNAi}$ at 25°C |
|---|---|---|---|
| *tubulin-GAL4* | Constitutive-ubiquitous | Embryonic lethal | Larval lethal (third instar) |
| *actin-GAL4* | Constitutive-ubiquitous | Embryonic lethal | Larval lethal (third instar) |
| *elav-GAL4* | Pan-neuronal | No locom. phenotype | NT |
| *nrv-GAL4* | Pan-neuronal specific in CNS and PNS | No locom. phenotype | NT |
| *D42-GAL4* | Motor neurons | No locom. phenotype | NT |
| *n-syb-GAL4* | Pan-neuronal | No locom. phenotype | NT |
| *repo-GAL4* | Glia | No locom. phenotype | NT |
| *69B-GAL4* | Embryonic epiderm, CNS, and imaginal discs | Larval lethal (first instar) | Larval lethal. Escapers with locom. defects at 18°C |
| *mhc-GAL4* | Myosin heavy chain promoter | Reduced climbing activity | NT |
| *Mef2-GAL4* | Somatic muscle cells, embryonic mesoderm (stages 10–17), embryonic cardioblast | Embryonic lethal | Semi-lethal (pupal stage) Escapers with locom. defects |
| *c179-GAL4* | Embryonic mesoderm and larval muscles | Reduced larval activity | Pupal lethal |
| *how*$^{24B}$*-GAL4* | Embryonic mesoderm. Precursors of the somatic muscles, visceral muscles, and cardiac cells. Larval nuclei of muscle fibers. | Reduced larval activity | Pupal lethal |
| *GMR-GAL4* | Eye imaginal disc | Retinal degeneration | NT |
| *nub-GAL4* | Wing imaginal disc | Wing size reduction and lost of patterning elements | Wing size reduction |
| *5053* GAL4 | Embryonic longitudinal visceral muscle founder cells | Vital and no adult locomotor phenotype | NT |
| *sr*$^{md710}$*-GAL4* | Embryonic and larval tendon cells. No expression in muscle or muscle precursors | Vital and no adult locomotor phenotype | NT |

1). Conversely, the expression of 2XUAS*dCNBP*[RNAi] driven by the multi-tissue *69B-GAL4* or the muscle-specific *myocyte enhancer factor 2 (Mef2)-GAL4* drivers caused early lethality at 29℃ (*Table 1*). At 25℃ or 18℃, lethality was reduced, allowing phenotypic analysis of adult 'escapers'. Adult flies carrying 2XUAS-*dCNBP*[RNAi] and either *Mef2-GAL4* or *69B-GAL4* showed strong locomotion defects, as denoted by the strong reduction of speed (0.01 *versus* 0.09 cm/s) and distance covered in 1 min (0.67 *versus* 8 cm) observed in silenced flies compared to controls (*Figure 1A*; *Figure 1—video 1*, *2*, *3*; *Gulyás et al., 2016*).

To study the effects of *CNBP* knockdown in differentiated muscle, we utilized an *mhc*-GAL4 driver, which induces the expression of transgenes later during muscle development, compared to *Mef2-GAL4*. Although these flies were viable, they showed defective locomotion, indicating that the integrity of *dCNBP* expression is required for locomotor activity also at later stages of muscle development. Wild-type flies usually display a strong negative geotactic response: when tapped to the bottom of a vial they rapidly run to the top. As they get older or manifest locomotion dysfunction, flies no longer climb to the top of the vial, but make short abortive climbs and fall back to the bottom. The climbing activity of *mhcGAL4>*2XUAS*dCNBP*[RNAi] flies was measured using the *Drosophila* activity monitoring (DAM) system (TriKinetics Inc, Waltham, MA; see Materials and methods), which allows to quantify fly locomotion capabilities based on their negative geotactic response. As shown in *Figure 1B*, *mhcGAL4>*2XUAS*dCNBP*[RNAi] flies exhibited a strong reduction (~50%) in the number of climbing events performed in 30 min compared to control flies.

We also investigated the consequences of *dCNBP* silencing in the embryonic mesoderm and larval muscles. RNAi constructs expressed under the control of the *c179*- or *how*[24B]-GAL4 drivers (*Table 1*) caused late pupal lethality at both 25℃ and 29℃ (*Table 1*). We thus analyzed the activity of RNAi-expressing larvae by measuring the numbers of peristaltic waves performed in 1 min. As shown in *Figure 1C*, both drivers caused a significant reduction of peristaltic waves compared to controls.

To further validate our findings and exclude that the phenotype could be linked to non-specific RNAi effects, we turned to an on-locus loss-of-function allele (dubbed *dCNBP*[k]). *dCNBP*[k] carries a P element insertion in the *dCNBP* locus (*CG3800*) causing lethality when homozygous (larvae die at the second instar). *dCNBP*[k] mutant larvae were examined for the expression of the dCNBP by immunoblotting and for their locomotor phenotype. We found that in these larvae the *dCNBP* product is completely absent compared to wild type (*Figure 1E*). We analyzed the locomotion activity of *dCNBP*[k] mutant larvae by measuring the numbers of peristaltic waves/min. As shown in *Figure 1D*, we found that, consistent with the RNAi data, *dCNBP* mutant larvae displayed a significant reduction of peristaltic waves compared to wild-type controls.

To further confirm that locomotor defects are a specific consequence of dCNBP depletion, we generated transgenic flies bearing an RNAi-resistant cDNA (UAS*dCNBP-3HAres*) which contains appropriate synonymous substitutions in the *dCNBP* coding sequence to be resistant to RNAi-mediated degradation. Expression of this construct under the control of the *c179-GAL4* driver rescued larval dCNBP loss-dependent locomotor phenotype (*Figure 2A*), confirming that this phenotype is specifically caused by *dCNBP* depletion.

We next investigated whether the human CNBP (hCNBP) orthologue could functionally rescue the prominent *dCNBP* phenotype, by expressing a *UAS hCNBP-FLAG* construct in the *Drosophila* muscle by using the *c179-GAL4* driver. As shown in *Figure 2A*, hCNBP reversed the locomotion defects of the *dCNBP*[RNAi]-depleted larvae (*Figure 2A*), indicating that hCNBP locomotor function is evolutionarily conserved from fly to human. The expression level of both transgenes were verified and quantified by western blot (*Figure 2B*).

## The ODC-polyamine pathway is involved in CNBP loss-of-function locomotor phenotype and is downregulated in muscles from DM2 patients

Having found that the absence of dCNBP in muscle tissues causes significant locomotor defects, we next sought to identify the CNBP-regulated mechanisms responsible for the observed phenotype.

Since we had previously found that mammalian CNBP regulates polyamine metabolism by affecting translation of ODC in cancer cells (*D'Amico et al., 2015*; *Sammons et al., 2011*; *Benhalevy et al., 2017*), we asked if a similar mechanism could play a role in this context.

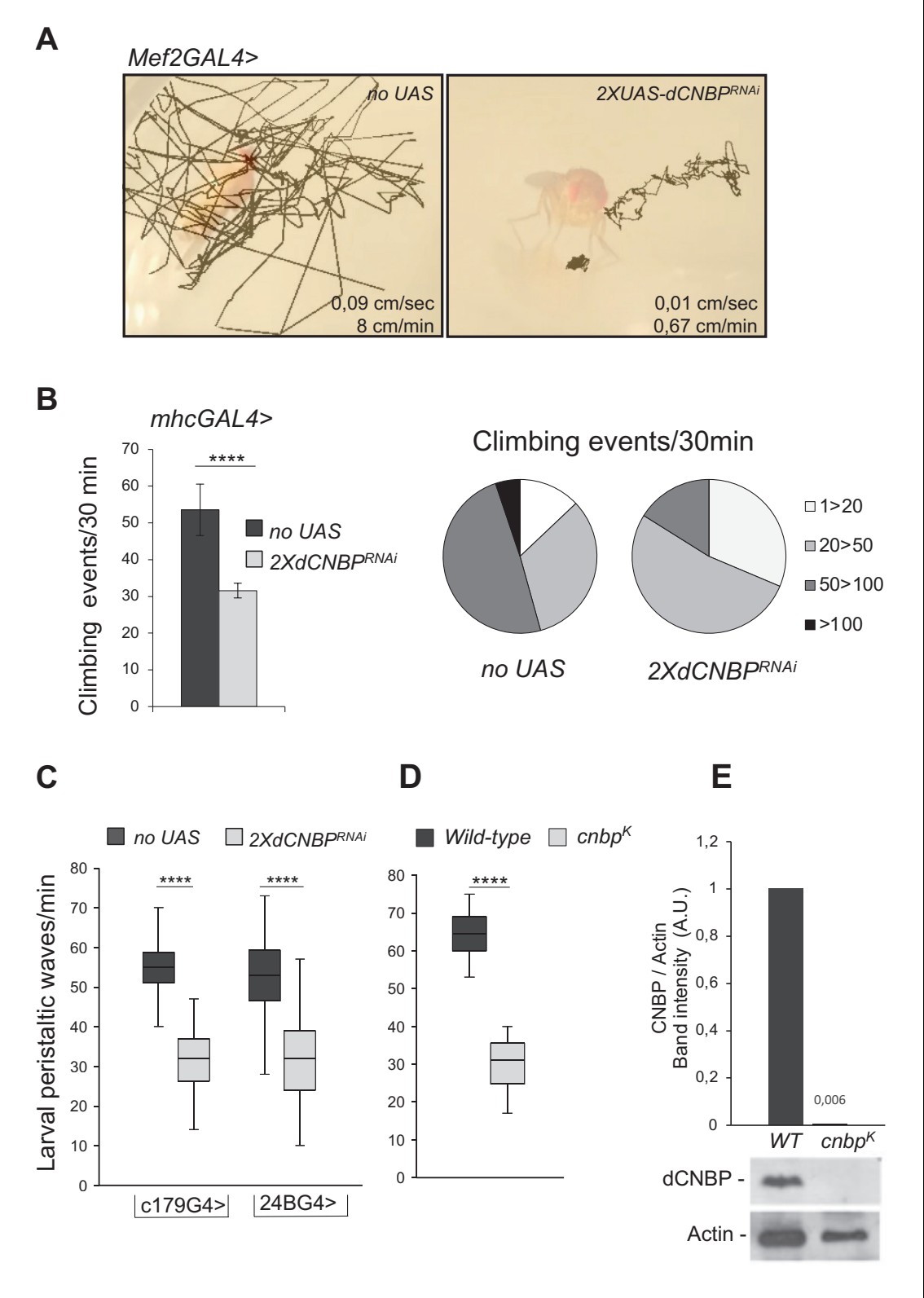

**Figure 1.** Specific *dCNBP* depletion in muscle tissues results in locomotor defects. (**A**) Locomotion activity in escapers adult flies expressing (**B**) *UAS-dCNBP^RNAi-16283*; *UAS-dCNBP^RNAi-16284* (2X*dCNBP^RNAi*) driven by the *myocyte enhancer factor 2 (Mef2)-GAL4* at 25˚C, compared to control flies (*no UAS*). Fly movements were recorded (*Figure 1—video 1*, *2*, *3*) and then analyzed by the animal tracker ImageJ plugin, to quantify both fly speed (average, cm/s) and distance covered in 1 min (*Gulyás et al., 2016*). (**C**) Climbing activity defects in adult flies expressing 2X*dCNBP^RNAi* driven by the

*Figure 1 continued on next page*

*Figure 1 continued*

*mhc-GAL4* at 29°C. The locomotion activity was measured by the *Drosophila* activity monitoring (DAM) system, as the number of climbing events in 30 min $\geq$ 80 males tested for each genotype. On the left, climbing performance of control flies (*no UAS*) or *dCNBP*-depleted flies (*2XdCNBP^RNAi*) 7 days after eclosion represented as the average of climbing events (CEs) in 30 min (error bars represent SEM; ****p<0.0001, Mann-Whitney-Wilcoxon test). On the right, quantitative grouping of climbing performances in four different classes. Classes 1–20 (white area) and 20–50 (light gray area) CEs are highly represented in RNAi flies (*2XdCNBP^RNAi*), while classes 50–100 CEs (dark gray area) are more frequent in control flies (*no UAS*). Only control flies have the ability to perform more than 100 CEs in 30 min (black area). Full data in **Figure 1—source data 1**. (D) Box plot representation of the distribution of peristaltic contraction rates performed in 1 min by control (*no UAS*) or *UAS-dCNBP^RNAi-16283*; *UAS-dCNBP^RNAi-16284* (*2XdCNBP^RNAi*) third instar larvae under the control of either *c179GAL4* or *24BGAL4* driver at 25°C (****p<0.0001, t-test); $\geq$10 larvae tested for each genotype in at least three independent experiments. Full data in **Figure 1—source data 1**. (E) Box plot representation of the distribution of peristaltic contraction rates performed by control (*wild-type [WT]*) or *cnbp^k* mutant second instar larvae in 1 min; $\geq$30 larvae tested for each genotype (p < 0.0001, Mann-Whitney-Wilcoxon test). In (C and D) the line inside the box indicates the median for each genotype and box boundaries represent the first and third quartiles; whiskers are min and max in the 1.5 interquartile range. Full data in **Figure 1—source data 1**. (F) Immunoblot showing the levels of dCNBP in extract obtained from *cnbp^k* mutant second instar or from WT control larvae with the corresponding band quantification normalized on the loading control (IMAGE J 1.50i; quantification data in source data for western blot [WB] quantification). Actin, loading control. A.U., arbitrary unit.

The online version of this article includes the following video, source data, and figure supplement(s) for figure 1:

**Source data 1.** Adult and larval movement measurements as shown in **Figure 1B-D**.

**Figure supplement 1.** dCNBP knockdown efficacy in absence of phenotype.

**Figure 1—video 1.** Example of locomotor movement of a control adult fly (*myocyte enhancer factor 2 [Mef2]-GAL4 > no UAS at 25°C*).

https://elifesciences.org/articles/69269#fig1video1

**Figure 1—video 2.** Example of locomotor defects of a dCNBP-interfered adult fly (*myocyte enhancer factor 2 [Mef2]-GAL4 > UAS-dCNBP^RNAi-16283*; *UAS-dCNBP^RNAi-16284* at 25°C).

https://elifesciences.org/articles/69269#fig1video2

**Figure 1—video 3.** Second example of locomotor defects of a dCNBP-interfered adult fly (myocyte enhancer factor 2 [Mef2]-GAL4>UAS-dCNBPRNAi-16283; UAS-dCNBPRNAi-16284at25°C).

https://elifesciences.org/articles/69269#fig1video3

We performed western blotting analysis of *dCNBP*-deficient larvae and observed that the levels of Odc are significantly reduced in larvae lacking dCNBP (in both *dCNBP* RNAi-expressing and *cnbp^k* mutant larvae, **Figure 3A** and **Figure 3—figure supplement 1**, respectively), compared to wild-type controls. Accordingly, the content of putrescine, the downstream product of Odc enzymatic activity, was strongly reduced (**Figure 3B** and **Figure 3—figure supplement 1**), confirming that dCNBP also regulates Odc and polyamine levels in flies.

To determine whether the locomotion defect caused by CNBP deficiency is linked to this mechanism, we analyzed the consequences of *dOdc* loss-of-function. We achieved an RNAi-mediated repression of both fly *Odc* genes, singularly or together (*dOdc1* and *dOdc2*; **Rom and Kahana, 1993**), by crossing RNAi lines (VDRC 30039 and 104597) to the *c179GAL4* driver (**Figure 4A**). *Odc* depletion caused adult lethality while in larvae displayed a significant impairment of peristaltic waves linked to larval locomotion activity and reduction of putrescine levels (**Figure 4—figure supplement 1**). Interestingly, putrescine reduction was significant in both *dOdc1* and *dOdc2* single or double knockdown, but not additive. This latter observation was likely attributable to the activation of compensatory mechanisms, such as the increased uptake of extracellular polyamines through specific transporters and/or intracellular polyamine interconversion (**Casero et al., 2018**), which prevent the total depletion of intracellular polyamines even in case of complete ODC blockade.

Similarly, d*Odc1^MI10996* (Bloomington #56103) mutant flies displayed significant locomotor defects associated with polyamine decrease (**Figure 4B**), and feeding of wild-type flies with DFMO, an irreversible ODC inhibitor, caused a significant reduction of larval motility (**Figure 4C**) demonstrating that genetic or pharmacological inhibition of dOdc strongly phenocopies dCNBP downregulation effects.

To determine if the ODC-polyamine axis is impaired also in the human disease, we studied muscle biopsies obtained from DM2 patients compared to those from healthy individuals. Remarkably, immunoblot analysis showed that the levels of both CNBP and ODC were reduced in DM2 patients compared to controls (**Figure 5A**). Consistently, we found that the content of the ODC metabolite putrescine was also significantly reduced in DM2 patients, thus indicating that polyamine synthesis might indeed be downregulated in these patients (**Figure 5B**).

In contrast, the levels of CNBP were not altered in a transgenic fly model of DM2 that ectopically expresses pure, uninterrupted CCUG-repeat expansions ranging from 200 to 575 repeats in length (BDSC 79583-79584-79585) and recapitulates some key features of human DM2 including RNA repeat-induced toxicity, ribonuclear foci formation, and changes in alternative splicing (*Yu et al., 2015*; *Figure 5—figure supplement 1*). These results suggest that the observed CNBP downregulation in DM2 patients is not due to the toxic RNA accumulation, but is rather a consequence of a different mechanism, specifically related to the intronic alteration. Unfortunately, the limited amount of patient samples did not allow us to investigate the mechanism underlying this downregulation. Further studies with muscle samples from patients will be required to elucidate this issue.

## dCNBP regulates dOdc translation

We next investigated the molecular mechanisms through which dCNBP controls *dOdc* expression and consequently polyamine metabolism.

We did not observe any significant reduction of *dOdc* mRNA levels in dCNBP RNAi-depleted muscles (*Figure 6A*), indicating that CNBP does not regulate *dOdc* mRNA synthesis or stability but rather its protein levels. In this regard, since our previous data in mammalian cells indicated that CNBP regulates IRES-dependent translation of ODC (*D'Amico et al., 2015*), we wondered if this mechanism might also be operating in flies. To this end we cloned the 5'UTR of *dOdc1* into a bicistronic renilla-luciferase reporter vector, which allows detection of IRES activity and tested the ability of dCNBP to induce *dOdc1* IRES-mediated translation. However, ectopic expression of dCNBP did not result in any significant change of reporter activity in mammalian cells, while it significantly

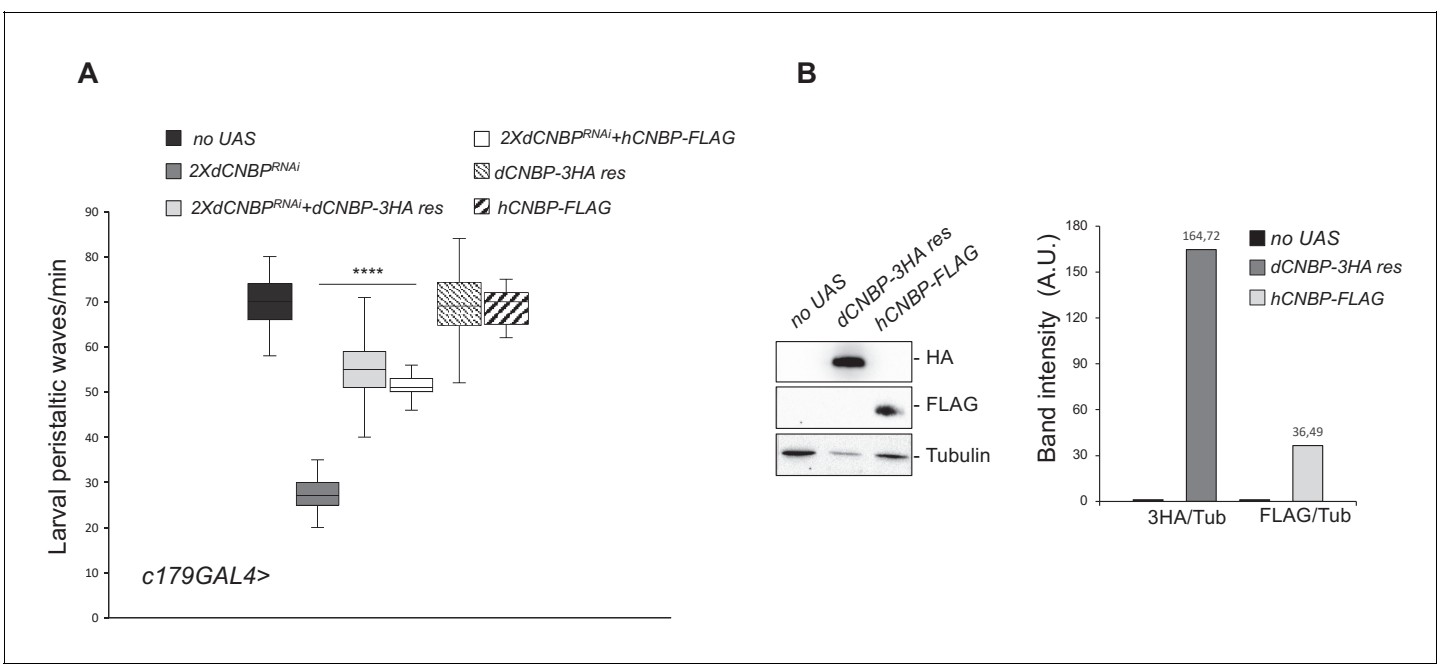

**Figure 2.** Cellular nucleic acid-binding protein (CNBP) overexpression rescues the locomotion phenotype induced by muscular *dCNBP* depletion. *dCNBP* knockdown in embryonic mesoderm causes a significant reduction of larval peristaltic waves rescued by the expression of either *dCNBP* or *hCNBP* transgenes (25°C). (A) Box plot representation of the distribution of peristaltic contraction rates performed by third instar larvae of the following genotypes: only *c179GAL4* driver (*no UAS*), *c179GAL4>UAS-dCNBP^{RNAi-16283}*; *UAS-dCNBP^{RNAi-16284}* (2X*dCNBP^{RNAi}*), *c179GAL4>2XdCNBP^{RNAi}* + *UAS-dCNBP-3HA-res* (a *dCNBP-3HA* transgene resistant to *2XUASdCNBP*-induced RNAi), *c179GAL4>2XdCNBP^{RNAi}* + UAS-*hCNBP-FLAG*. The line inside the box indicates the median for each genotype and box boundaries represent the first and third quartiles; whiskers are min and max in the 1.5 interquartile range (****p<0.0001, Kruskal-Wallis with post hoc Dunn's test); ≥10 larvae tested for each genotype in at least three independent experiments. Full data in *Figure 2—source data 1*. (B) The expression levels of both *UAS-dCNBP-3HA-res* and UAS-*hCNBP-FLAG* were analyzed by immunoblotting using antibodies against either the HA or the FLAG tag, compared to controls (*no UAS*). Bands were quantified by IMAGE J 1.50i and normalized on the loading control (quantification data in source data for western blot [WB] quantification). Tubulin, loading control. A.U., arbitrary unit. The online version of this article includes the following source data for figure 2:

**Source data 1.** Larval movement measurements as shown in *Figure 2A*.

induced the activity of a bicistronic vector containing the 5'UTR of human ODC (hODC), thus excluding that dCNBP could regulate *dOdc* translation through an IRES-mediated mechanism (*Figure 6—figure supplement 1*).

In a previous report it was shown that mammalian CNBP regulates translation of several target mRNAs via an association with G-rich recognition elements (RRE), thereby resolving their G4 stable structures and promoting translational elongation (*Benhalevy et al., 2017*). Interestingly, one of the targets identified in that study was the mRNA of ODC and our in silico analysis by RBPmap (*Paz et al., 2014*; http://rbpmap.technion.ac.il) predicted the presence of several UGGAGNW motifs (the most common RRE bound by hCNBP; *Figure 6—figure supplement 2*) in the *Drosophila Odc1* coding sequence. Thus, we tested if dCNBP regulates translational efficiency of *dOdc1* by binding its mature mRNA. To this end, we performed RNA immunoprecipitation (RIP) assay on S2 insect cell extracts and found that CNBP efficiently binds *Odc* mRNA (*Figure 6B*, left). In addition, dCNBP was efficiently associated with dOdc mRNA in control (no UAS) but not in CNBP-deficient larval muscles (*Figure 6B*, right). The RNAi efficiency was confirmed by western blotting (*Figure 6B* right and *Figure 6—figure supplement 3A*), demonstrating the specificity of the binding. Furthermore, in a heterologous system, after transfection of a vector expressing the *dOdc1* CDS, but lacking its UTRs, Odc protein synthesis was downregulated by CNBP depletion, while the mRNA levels remained unchanged (*Figure 6—figure supplement 4*), thus supporting the hypothesis that dCNBP regulates *dOdc* mRNA translation by acting on its coding region.

To determine whether dCNBP influences translation of *dOdc* mRNA, we performed a sucrose fractionation of cytoplasmic lysates obtained from S2 cells in which *dCNBP* mRNA was knocked

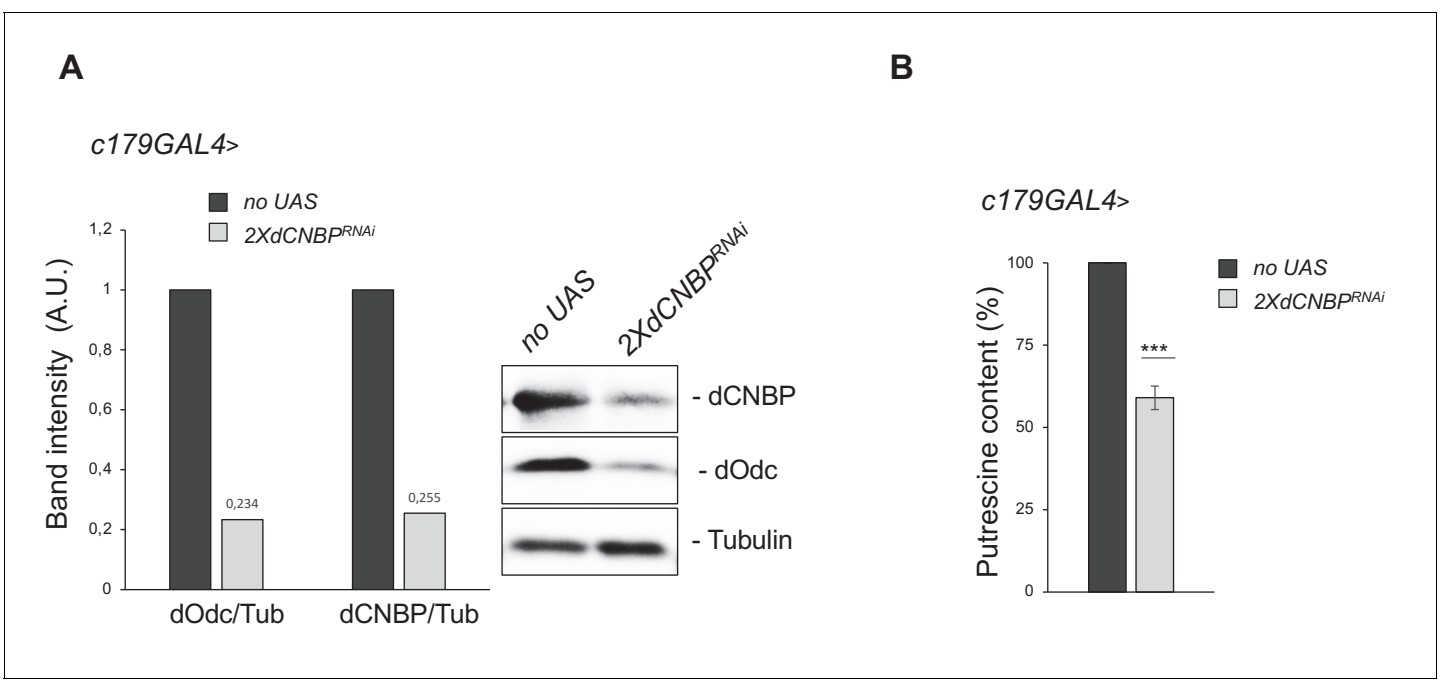

**Figure 3.** CNBP regulates the ornithine decarboxylase (ODC)/polyamine axis. Levels of both Odc and putrescine are significantly reduced in *dCNBP*-depleted larvae compared to wild-type controls. (**A**) Immunoblot showing the levels of both dCNBP and dOdc in extract obtained from *tubGAL4>2XdCNBP^RNAi* third instar larvae compared to control (*no UAS*), with the corresponding band quantification normalized on the loading control (IMAGE J 1.50i; quantification data in source data for western blot [WB] quantification). Actin, loading control. A.U., arbitrary unit. (**B**) Columns represent the fold difference of putrescine content in third instar larvae bearing the *c179GAL4* driver alone (*no UAS*) or in combination with double copy dCNBP RNAi-expressing larvae (*UAS-dCNBP^RNAi-16283*; *UAS-dCNBP^RNAi-16284*, named 2X*dCNBP^RNAi*). Error bars represent SEM; \*\*\*p>0.001, \*\* p>0.002, in unpaired t-test. A pool of 10 larvae has been tested for each genotype in three independent experiments. Full data in *Figure 3—source data 1*.

The online version of this article includes the following source data and figure supplement(s) for figure 3:

**Source data 1.** Putrescine content quantification as shown in *Figure 3B* and *Figure 3—figure supplement 1B*.

**Figure supplement 1.** Larval locomotor defect observed in *cnbp^k* mutants correlates with the reduction of ornithine decarboxylase (Odc) protein and polyamine levels.

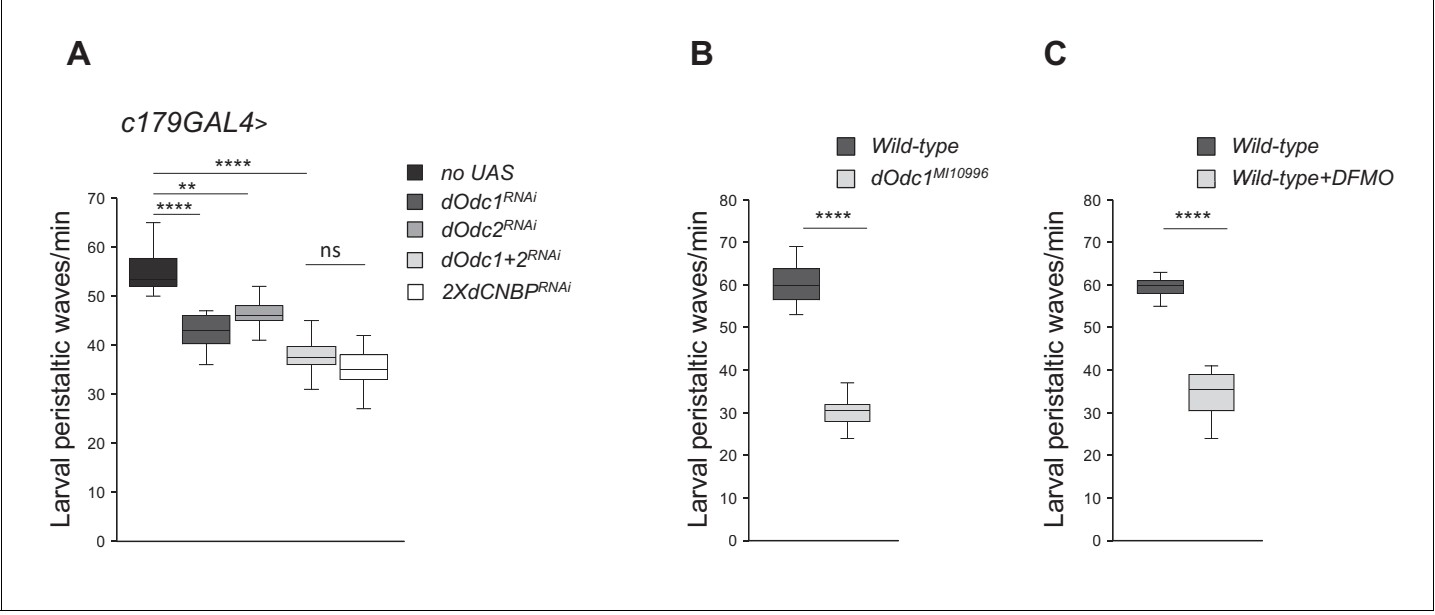

**Figure 4.** Ornithine decarboxylase (Odc) depletion phenocopies the *dCNBP* locomotor defects. Box plot representation of the distribution of peristaltic contraction rates performed by third instar larvae of the reported genotypes in 1 min. (**A**) *c179GAL4>no UAS, UAS-Odc1$^{RNAi-30039}$, UAS-Odc2$^{RNAi-10459}$, UAS-Odc2$^{RNAi-10459}$; UAS-Odc1$^{RNAi-30039}$, or UAS-dCNBP$^{RNAi-16283}$; UAS-dCNBP$^{RNAi-16284}$*. In the graph legend UAS in transgenic RNAi lines is omitted for simplicity. (**B**) Controls (*wild type*) and *dOdc1$^{MI10996}$* mutant larvae. (**C**) Controls fed with standard fly food (*wild type*) or after DFMO treatment (5 mM/day; *wild type + DFMO*). The line inside the box indicates the median for each genotype and box boundaries represent the first and third quartiles; whiskers are min and max in the 1.5 interquartile range (**p<0.001; ****p<0.0001; ns, not significant, Kruskal-Wallis with post hoc Dunn's test for multiple comparison or Mann-Whitney-Wilcoxon test for); ≥10 larvae tested for each genotype in at least three independent experiments. All full data in *Figure 4—source data 1*.

The online version of this article includes the following source data and figure supplement(s) for figure 4:

**Source data 1.** Larval movement measurements as shown in *Figure 4A-C* and putrescine content quantification as shown in *Figure 4—figure supplement 1*.

**Figure supplement 1.** Larval locomotor defect observed as a consequence of ornithine decarboxylase (Odc) depletion correlates with the reduction of polyamine levels.

---

down or from control cells (*Figure 6C and D*). RNA was extracted from each fraction and analyzed by qRT-PCR (*Figure 6D*). The RNAi efficiency was confirmed by western blotting (*Figure 6D* right and *Figure 6—figure supplement 3B*). In line with our assumption, in control lysates we found significant levels of *dOdc* mRNA in polysome fractions, the same where dCNBP was detected and co-purified with the ribosomal protein RpS6 (*Antonucci et al., 2014*), indicating that Odc is actively translated. In contrast, the levels of *dOdc* mRNA were strongly reduced in the polysome fraction of dCNBP-deficient S2 cells, while they were significantly increased in the non-translating fractions (60–40S and free mRNA), demonstrating that dCNBP is required for *dOdc* mRNA loading into polysomes and therefore for its active translation (*Figure 6C and D*).

## Restoration of polyamine metabolism in dCNBP-deficient flies rescues locomotor phenotypes

To verify that the above-described mechanism leading to alteration of polyamine metabolism is truly responsible for the observed locomotor phenotype, we performed rescue experiments. We first fed *dCNBP* mutant or RNAi-expressing flies with putrescine dissolved into their food. As shown in *Figure 7A and B*, RNAi-expressing or *dCNBP* mutant larvae substantially recovered the locomotor defects after putrescine administration and, as expected, putrescine content significantly increased compared to controls reared on standard food (*Figure 7—figure supplement 1A*). Similar data were obtained with spermidine (*Figure 7—figure supplement 1B*). Then, we used the *Mef2-GAL4* or the *c179-GAL4* lines to drive simultaneous expression of *UAS-dOdc1* (*Gupta et al., 2013*) and 2XUAS-*dCNBP$^{RNAi}$*, and found that dOdc1 reconstitution significantly ameliorates the locomotor

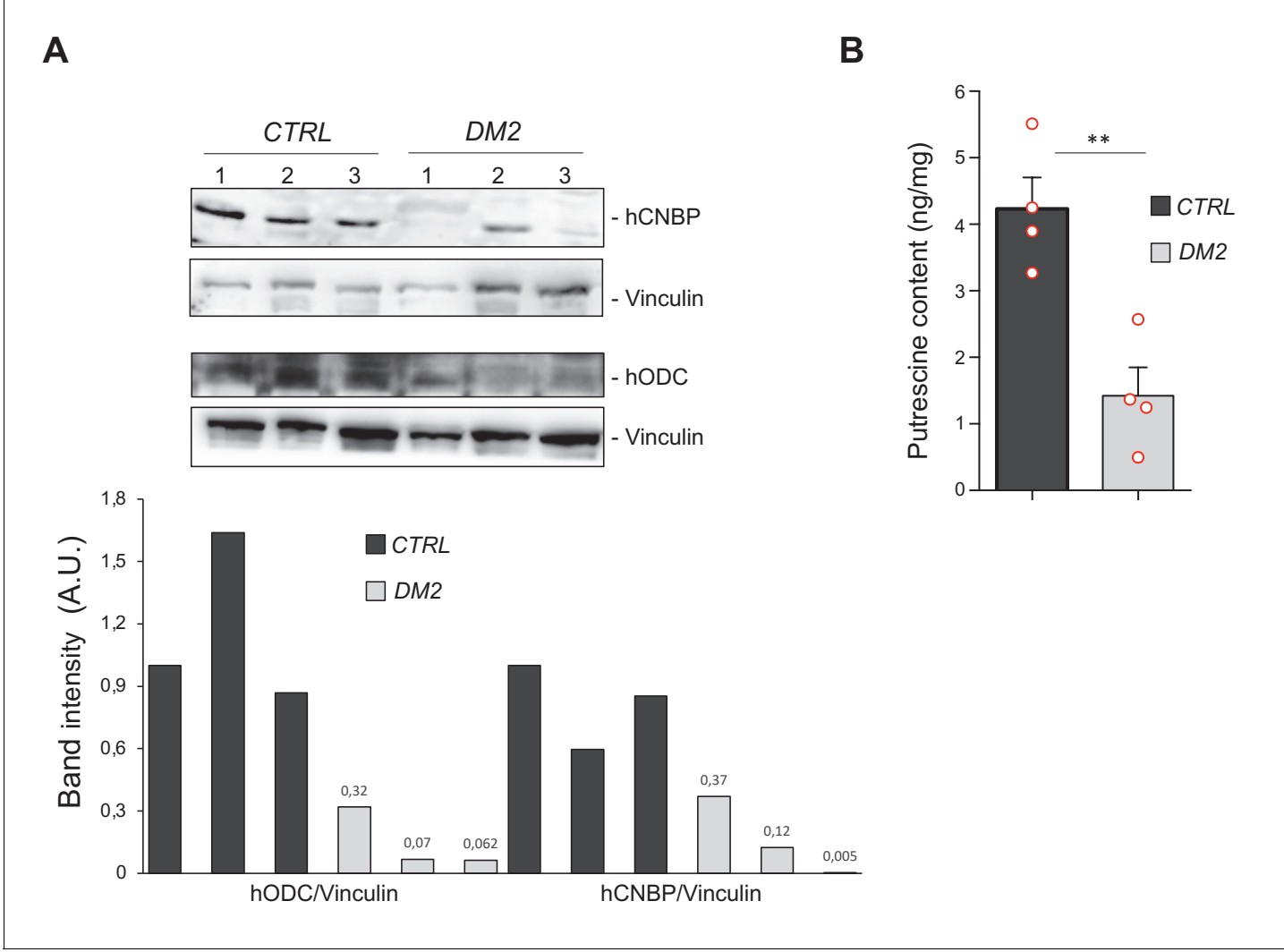

**Figure 5.** Polyamine metabolism is impaired also in myotonic dystrophy type 2 (DM2) muscles. Cellular nucleic acid-binding protein (CNBP) and ornithine decarboxylase (ODC) content correlates with polyamine levels in muscle cells from DM2 patients. (**A**) Immunoblot showing the levels of both human CNBP (hCNBP) and human ODC (hODC) proteins in three DM2 or control muscle cells, with the corresponding band quantification normalized on the loading control (IMAGE J 1.50i; quantification data in source data for western blot [WB] quantification). Vinculin, loading control. A.U., arbitrary unit. (**B**) Columns represent putrescine content in muscle cells obtained from four DM2 patients (*CTRL*) or from four healthy individuals (*DM2*), expressed in ng/mg of tissue. Error bars represent SEM; **p>0.001, in unpaired t-test. Full data in *Figure 5—source data 1*.

The online version of this article includes the following source data and figure supplement(s) for figure 5:

**Source data 1.** Putrescine content quantification as shown in *Figure 5B*.

**Figure supplement 1.** Expression levels of dCNBP are not affected by the expression of CCUG-expanded repeat RNA.

phenotype in *dCNBP*-depleted larvae (*Figure 7C*). The levels of the CNBP in these flies were very low and comparable to those observed in lines lacking only *dCNBP* (*Figure 7—figure supplement 2* and *Figure 3A*). This indicates that the rescue of the locomotor defect can only be attributed to overexpression of *dOdc1*, and not to a potential dilution of the GAL4 driver.

Of note, we showed that feeding with 1 mM putrescine mutants for *dystrophin* (*Dys^det-1*), a fly model of Duchenne muscle dystrophy (DMD), did not recover the *Dys*-dependent larval locomotor abnormalities (*Figure 7D*), indicating that the recovery of polyamine metabolism is specifically required for alleviating *dCNBP* loss-of-function locomotor defects.

It is known that polyamines decrease during aging in *Drosophila* (*Gupta et al., 2013*). Moreover, the decline of locomotor ability with age is common in many species of animals and muscular dystrophies gradually progress with age, along with increased muscle breakdown. Thus, we performed the

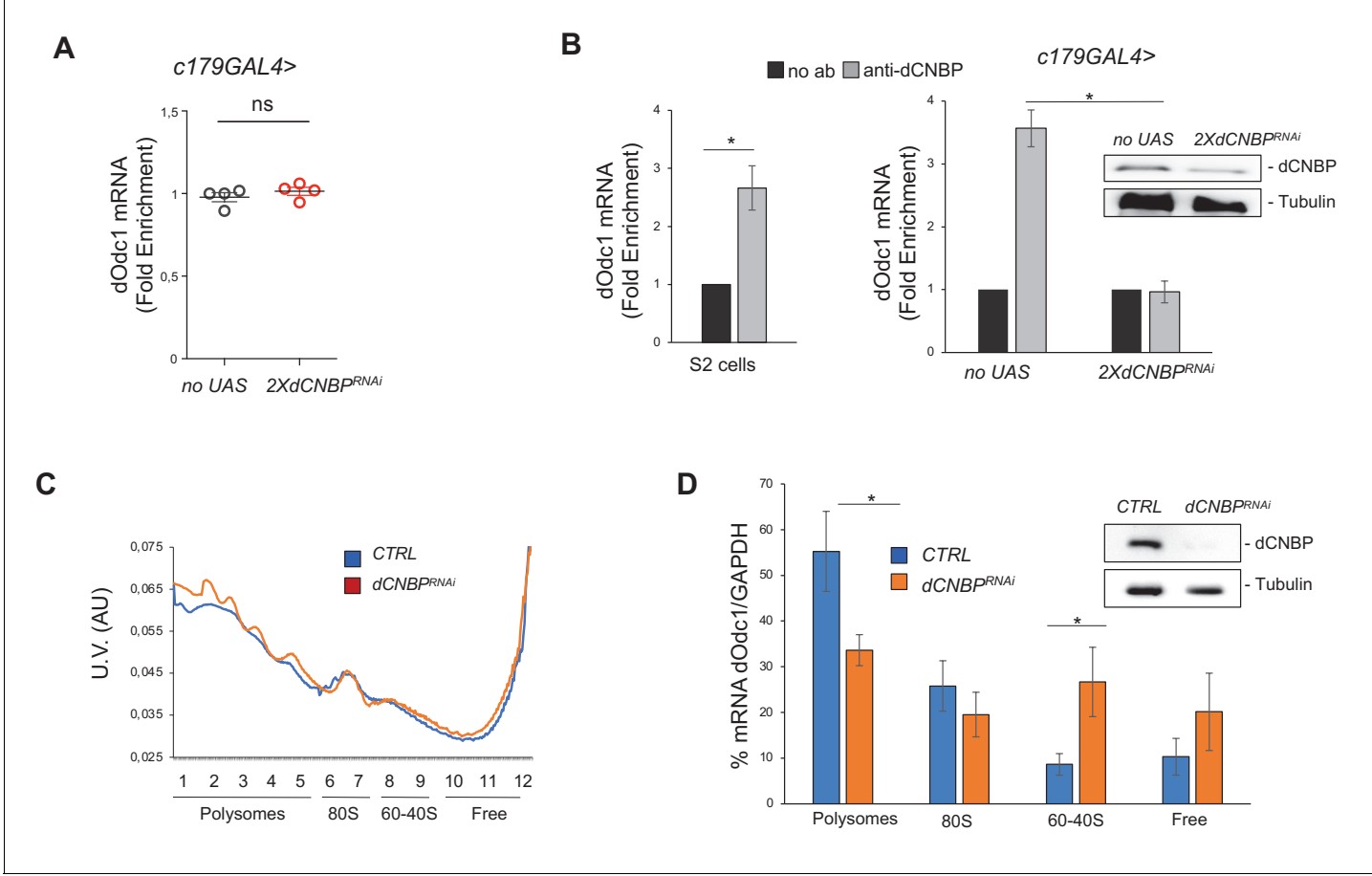

**Figure 6.** dCNBP controls polyamine metabolism through the binding and the translational control of *dOdc* mRNA. (**A**) *dOdc1* mRNA levels (qPCR), normalized with the housekeeping *RPL11* mRNA third instar larvae bearing *c179GAL4* driver alone (*no UAS*) or in combination with *UAS-dCNBP^RNAi-16283*; *UAS-dCNBP^RNAi-16284* (*2XdCNBP^RNAi*). ns, not significant in unpaired t-test. Dots correspond to four independent biological replicates; bars indicate the mean and SEM. (**B**) Cellular nucleic acid-binding protein (CNBP) binds *dOdc1* mRNA. qRT-PCR analysis on mRNAs immunoprecipitated by anti-dCNBP antibody or control IgG antisera in S2 cells extracts (left graph), or in dCNBP-depleted (*2XdCNBP^RNAi*) or not (*no UAS*) larval extracts (right graph). The results are indicated as fold difference, relative to IgG. Error bars represent SEM of three independent experiments; *p < 0.05, in t-test. The presence of dCNBP in *c179GAL4>2XUASdCNBP^RNAi* or control (*no UAS*) larval carcasses was analyzed by western blotting (right). Tubulin, loading control. (**C**) Representative polysome profiles (of at least three independent experiments) of dCNBP-deficient (*dCNBP^RNAi*) or control (*CTRL*) S2 cells. Cytoplasmic lysates were fractionated on 15–50% sucrose gradients. (**D**) qPCR analysis of *dOdc1* mRNA loaded in the different polysome fractions, GADPH was used to normalize the values. (*p < 0.05, t-test. Error bars represent SEM of experiments performed in quadruplicates and repeated at least three times.) The presence of dCNBP in interfered or not interfered S2 cells was analyzed by western blotting (right). Tubulin, loading control. All full data in *Figure 6—source data 1*.

The online version of this article includes the following source data and figure supplement(s) for figure 6:

**Source data 1.** Real-time qPCR data as shown in *Figure 6A, B, D*, in *Figure 6—figure supplement 1A-B* and in *Figure 6—figure supplement 4B*; polysome profile data as shown in *Figure 6C*.

**Figure supplement 1.** dCNBP does not control polyamine metabolism through *dOdc1* internal ribosome entry site (IRES)-dependent translation.

**Figure supplement 2.** In silico prediction of putative cellular nucleic acid-binding protein (CNBP) binding sites on the *dOdc1* mRNA by RBPmap.

**Figure supplement 2—source data 1.**

**Figure supplement 3.** Efficiency of dCNBP silencing.

**Figure supplement 4.** Cellular nucleic acid-binding protein (CNBP) promotes translation of *dOdc* mRNA.

DAM climbing assay with the *dCNBP* RNAi-expressing adults at different times after eclosion. We found that the *dCNBP*-depleted adults showed a faster ageing-dependent decline of climbing ability, as 15 days aged adult flies performed a significantly lower number of climbing events/30 min compared to wild-type control flies (*Figure 7—figure supplement 3*). Thus, it could be speculated that dCNBP depletion accelerates ageing-dependent locomotor decline, similar to that observed in

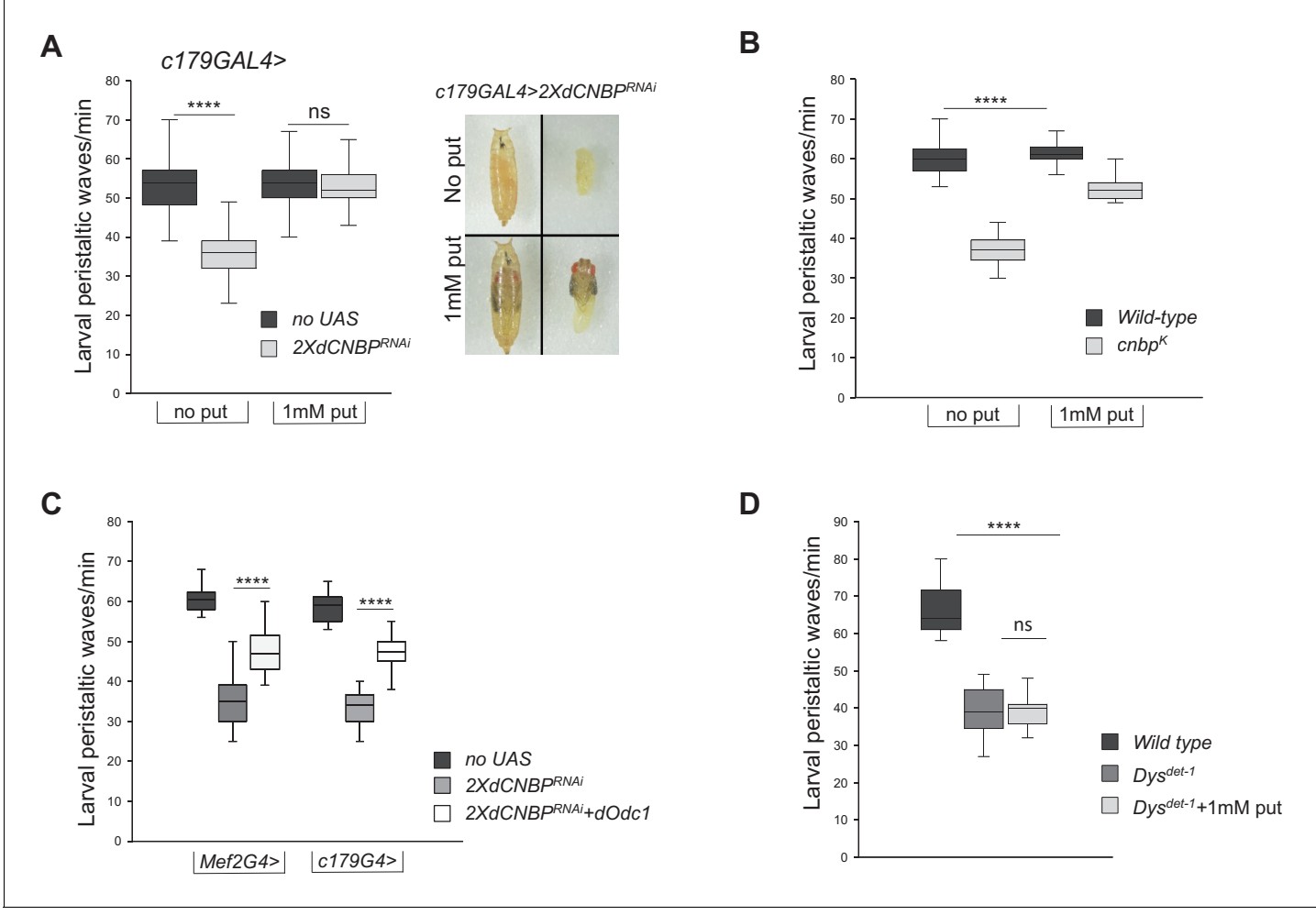

**Figure 7.** Odc and polyamine are responsible for the CNBP-dependent locomotor phenotype. (**A–B**) Rescue of locomotor phenotype in both *dCNBP*-depleted larvae (**A**) and *dCNBP* mutant larvae (**B**) by 1 mM putrescine feeding at 29°C. Box plot representation of the distribution of peristaltic contraction rates performed by the following genotypes: (**A**) *c179GAL4* driving *no UAS* or *UAS-dCNBP*[RNAi-16283]; *UAS-dCNBP*[RNAi-16284] (*2XdCNBP*[RNAi]) with (1 mM put) or without (no put) putrescine. Note how putrescine feeding of interfered individuals results also in a higher stage of pupal development with respect to individuals not treated (photo in A). (**B**) Control (*wild-type*) or *cnbp*[k] larvae with (1 mM put) or without (no put) putrescine. (****p<0.0001; ns, not significant, Kruskal-Wallis with post hoc Dunn's test). (**C**) Rescue of locomotor defects in *dCNBP*-depleted larvae by dOdc1 overexpression under the control of either *Mef2* or *c179GAL4* driver. Box plot representation of the distribution of peristaltic contraction rates performed by the following genotypes: *Mef2GAL4* or *c179GAL4* (*no UAS*), *Mef2GAL4* or *c179GAL4>2XdCNBP*[RNAi], *Mef2GAL4* or *c179GAL4>2XdCNBP*[RNAi] + dOdc1 (*UASdCNBP*[RNAi-16283]; *Mef2GAL4/UASdCNBP*[RNAi-16284]; *UASdOdc1/TM6B* or *UASdCNBP*[RNAi-16283]; *c179GAL4/ UASdCNBP*[RNAi-16284]; UAS d*Odc1/+*). (**A–B–C**) The line inside the box indicates the median for each genotype and box boundaries represent the first and third quartiles; whiskers are min and max in the 1.5 interquartile range (****p<0.0001; ns, not significant, Kruskal-Wallis with post hoc Dunn's test); ≥10 larvae tested for each genotype in at least three independent experiments. (**D**) Mutants for *dystrophin* (*Dys*[det-1]) present larval locomotor abnormalities that cannot be rescued by feeding larvae with 1 mM putrescine. Box plot representation of the distribution of peristaltic contraction rates performed by *Dys*[del-1] mutant larvae fed with or without putrescine (+1 mM put) with respect to *wild-type* control. The line inside the box indicates the median for each genotype and box boundaries represent the first and third quartiles; whiskers are min and max in the 1.5 interquartile range (ns, not significant, ****p<0.0001, Kruskal-Wallis with post hoc Dunn's test); ≥10 larvae tested for each genotype in at least two independent experiments. All full data in *Figure 7—source data 1*.

The online version of this article includes the following source data and figure supplement(s) for figure 7:

**Source data 1.** Larval and adult movement measurements as shown in *Figure 7A-D*, *Figure 7—figure supplement 1B* and *Figure 7—figure supplement 3*. Putrescine content measurment as shown in *Figure 7—figure supplement 1A*.

**Figure supplement 1.** Effects of other polyamines on the cellular nucleic acid-binding protein (CNBP)-dependent locomotor phenotype.

**Figure supplement 2.** dOdc1 overexpression does not affect dCNBP downregulation.

**Figure supplement 3.** *dCNBP*-depleted flies exhibited an ageing-dependent locomotor dysfunction.

**Figure supplement 4.** dCNBP depletion does not cause morphological changes of fly larval muscle tissues.

DM patients (*Mateos-Aierdi et al., 2015*). This acceleration may well be a consequence of an age-ing-dependent polyamines decrease (*Gupta et al., 2013*). However, we cannot also exclude that the ageing-dependent acceleration of the locomotor dysfunction upon CNBP silencing could be linked to either developmental defects or to heterogeneous backgrounds. Further studies will be required to address these points.

Together, these results indicate that ODC and polyamine defects are responsible for the observed CNBP loss-of-function locomotor phenotype.

## Discussion

It is widely recognized that DM1 and DM2 share many clinical features due to a common pathogenic mechanism, consisting in the toxic accumulation of RNA, resulting from the expansion of CTG trip-lets or CCTG quadruplets, respectively. However, the two diseases differ in some clinical manifesta-tions, such as the preeminent involvement of proximal muscles in DM2 and of distal muscles in DM1. Therefore, it is possible that additional mechanisms contribute to the pathogenesis of the two dis-eases, by acting as disease modifiers. Indeed, the pathologies of repeat expansion-associated dis-eases are very complex, as both coding and non-coding repeat expansions may involve a combination of mechanisms, including protein loss-of-function, toxic RNA gain-of-function, and toxic protein gain-of-function.

In DM2, the quadruplet expansion occurs within the first intron of the *CNBP* gene and this has given rise to the hypothesis that this genetic alteration may cause splicing defects/protein sequestra-tion, leading to reduced CNBP levels. However, while some studies reported that CNBP levels are significantly reduced in muscle of DM2 patients, other works failed to observe such a reduction (*Eisenberg et al., 2016*; *Eisenberg et al., 2009*; *Huichalaf et al., 2009*; *Raheem et al., 2010*; *Salisbury et al., 2009*; *Schneider-Gold and Timchenko, 2010*; *Wei et al., 2018*), most likely as a consequence of the limited sample sizes and the variability of the disease. Therefore, whether CNBP reduction plays a pathogenic role in DM2 is still a debated issue.

Previous studies in mice demonstrated that both heterozygous and homozygous deletion of CNBP alleles causes relevant muscle defects (*Chen et al., 2007*; *Wei et al., 2018*), suggesting a role of CNBP loss-of-function in the pathogenesis of the disease. In particular, while homozygous dele-tion of CNBP is associated to muscle atrophy and severe impairment of muscle performance at young age, the heterozygous CNBP KO mice show milder muscle dysfunctions, but develop a more pronounced locomotor phenotype at advanced age, reminiscent of DM2 disease (*Wei et al., 2018*). This latter observation is consistent with the onset of clinical manifestations in DM2 patients, which typically begins in the elderly, after the age of 60.

In the present work we have addressed for the first time the specific role of CNBP in muscle, using *D. melanogaster* as a model. Using muscle-specific drivers, expressed at various stages of mus-cle development, we have ablated *dCNBP* gene from muscle tissues and observed severe locomotor defects that, in analogy with observations in patients and other animal models, become more pro-nounced with age.

CNBP deficiency is sufficient to cause this effect as evidenced by the finding that reconstitution with either dCNBP or hCNBP fully rescues the locomotor phenotype.

We have found that when *dCNBP* is knocked down at early stages of muscle development, very severe phenotypes or lethality ensue, and that knock down specifically in differentiated muscles results in robust locomotor defects. This suggests that CNBP is necessary to ensure not only proper muscle development, but also its function in the adult. Surprisingly, in contrast with studies in homo-zygous KO mice showing marked muscle atrophy, our morphological analysis of muscle tissues did not show significant changes upon CNBP knockdown (*Figure 7—figure supplement 4*). A plausible explanation for this discrepancy could be that the phenotype observed in mice is the result of the constitutive loss of CNBP in all tissues, while in our models the protein was deleted exclusively in muscle territories, likely affecting their function but not their architecture. We did not see clear dif-ferences in muscle morphology also in dCNBP mutant larvae compared to controls (*Figure 7—fig-ure supplement 4*). However, dCNBP mutant animals die early during larval development (second instar), thus it is possible that such a short survival of dCNBP mutant larvae does not allow sufficient time for the muscle alterations to be fully developed and appreciated. Additionally, it cannot be

excluded that an early requirement of dCNBP for muscle development might be overshadowed by the presence of maternal contribution provided by the heterozygous mother flies.

Mechanistically, we provide evidence that the observed phenotype is linked to the ability of CNBP to control polyamine content, by regulating ODC translation. In our previous studies, we found that in mammalian cells, CNBP binds to the 5′UTR of ODC mRNA, thereby regulating IRES-mediated translation and polyamine metabolism (*D'Amico et al., 2015*). Indeed, mammalian ODC mRNA has a relatively long (about 350 nts) 5′UTR and its translation can be initiated at specific internal pyrimidine-rich sequences (*Pyronnet et al., 2000*) that were also found to bind CNBP (*Gerbasi and Link, 2007*). In contrast, the 5′UTR of dOdc measures only 27 nts, lacks the pyrimidine-rich sequences, and does not show any IRES activity after ectopic expression of CNBP. Therefore, dCNBP does not seem to regulate translation of dOdc through an IRES-mediated mechanism.

A previous work demonstrated that CNBP facilitates translational elongation in mammalian cells, by binding G-rich motifs and resolving stable secondary structures of a number of putative transcripts, being ODC mRNA among the targets identified in that screening, although not functionally validated (*Benhalevy et al., 2017*). These observations suggest a dual mode of CNBP regulation of ODC translation in mammals, at the level of both internal initiation and elongation across G-rich sites.

In this work we found that dCNBP binds *dOdc* mRNA and regulates its translation, likely acting at the coding region, thus supporting the conclusion that CNBP promotes *dOdc* translational elongation through the same mechanism described in mammalian cells and suggesting that the regulation of ODC by CNBP is a very important and evolutionary conserved mechanism.

Of note, in this work we have demonstrated that the locomotor defects caused by CNBP deficiency are linked to a significant decrease of polyamine content and, importantly, that the defects can be rescued by restoring dOdc expression or by polyamine supplementation. This molecular mechanism seems to be specifically linked to CNBP loss-dependent muscle dysfunction, as polyamine supplementation was unable to ameliorate the locomotor defects in a fly model of DMD. Our own data obtained from a small cohort of DM2 patients support the hypothesis that the polyamine metabolism is also altered in human DM2 muscle tissues. However, due to the heterogeneity of this disease, a study specifically addressing the representation of CNBP and polyamines, measuring their content in various muscles, as well as investigating the molecular mechanisms leading to CNBP downregulation in a large number of patients is needed to establish more compelling evidence. Thus, whether DM2 patients may benefit from polyamine supplementation represents a crucial question opened by this work that deserves further investigation. Interestingly, previous work demonstrated that reduced polyamine content correlates with the severity of muscle dysfunction in a mouse model of another form of human muscle dystrophy: LAMA2-congenital muscle dystrophy (CMD; *Kemaladewi et al., 2018*). Like DM2, CMD is characterized by phenotypic variability and differentially affects specific muscle groups, possibly as a consequence of a differential expression of polyamine regulators and polyamine content.

Therefore, it is possible that CNBP also acts as a disease modifier in DM2, causing the differential distribution of polyamine content among distal and proximal muscles, which in turn sustains the clinical heterogeneity of this disease.

How polyamines affect muscle function remains to be understood. A previous study reported that supplementation of both mice and *Drosophila* diet with spermidine extends their lifespan (*Eisenberg et al., 2016*; *Eisenberg et al., 2009*) and exerts protective effects on cardiac muscle of mice, by promoting cardiac autophagy, mitophagy, and mitochondrial respiration (*Eisenberg et al., 2016*; *Eisenberg et al., 2009*). Thus, it is possible that an impairment of these mechanisms in muscle may be involved in the observed phenotype of CNBP-deficient animals and possibly in DM2 patients. Moreover, polyamines are among the substances that have been reported to decline with age (*Gupta et al., 2013*; *Liu et al., 2008*) and the phenotype of CNBP-deficient animals or the clinical manifestation of DM2 patients is also correlated with the advanced age. Therefore, it is possible that polyamine may be involved, at least in part, in the ageing-dependent manifestations of the disease. Further studies on the role and mechanism of action of polyamines in muscle function are thus required to elucidate this critical issue.

In conclusion, we have identified an unprecedented mechanism whereby dCNBP controls muscle function by regulating the ODC/polyamine axis (*Figure 8*). This function of dCNBP we have

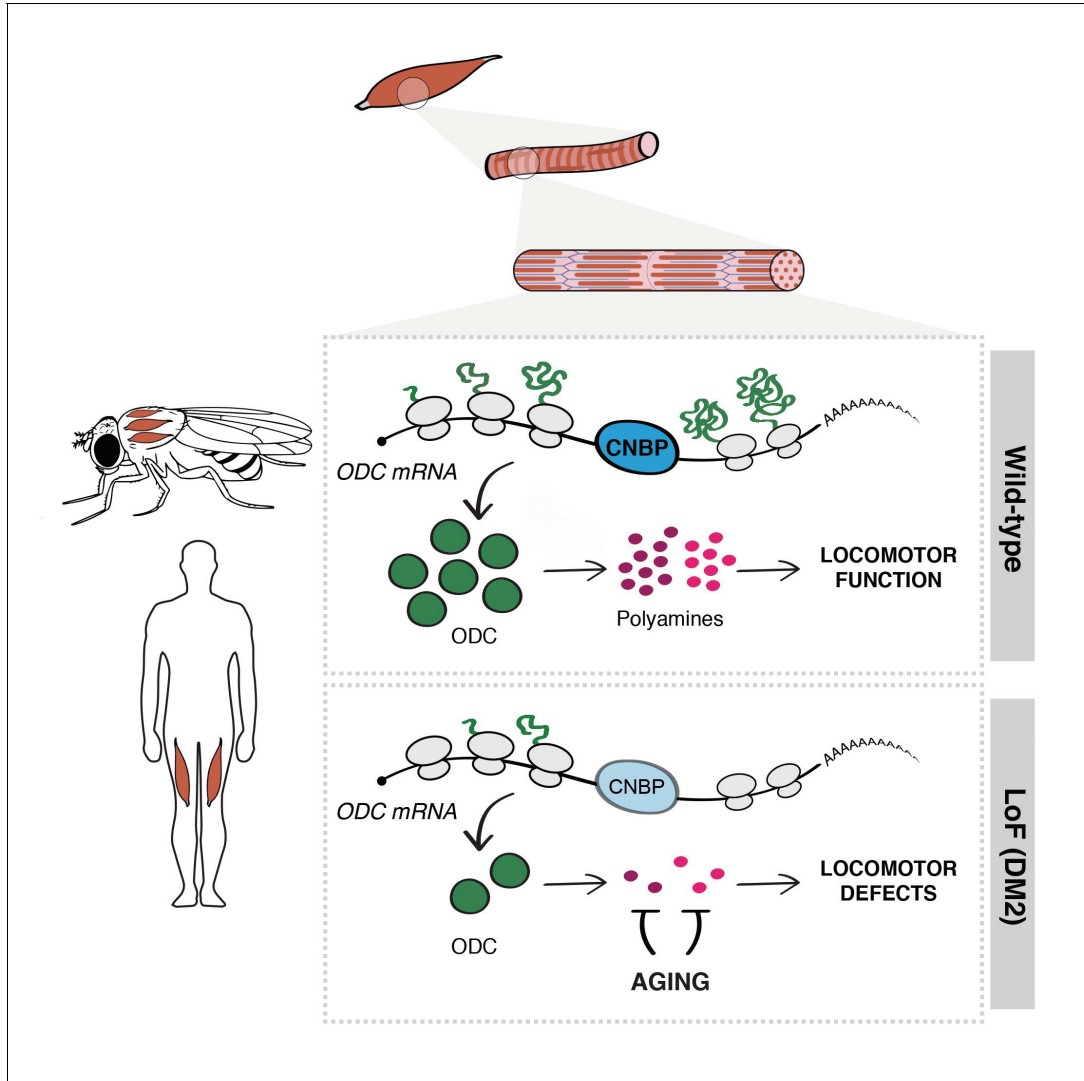

**Figure 8.** Cellular nucleic acid-binding protein (CNBP) controls muscle function by regulating polyamine metabolism. Schematic representation of the mechanism of action of CNBP on muscle function in both *Drosophila* and humans. CNBP binds ornithine decarboxylase (ODC) mRNA and regulates its translation, leading to increased polyamine content. Lack of CNBP impairs locomotor function through ODC-polyamine downregulation.

described in *Drosophila* seems to be evolutionarily conserved in vertebrates, with relevant implications in DM2 disease.

## Materials and methods

**Key resources table**

| Reagent type (species) or resource | Designation | Source or reference | Identifiers | Additional information |
|---|---|---|---|---|
| Genetic reagent (*Drosophila melanogaster*) | *actin-GAL4* | Bloomington | 25374 | *y[1] w[*]; P{Act5C-GAL4-w}E1/CyO* |
| Genetic reagent (*Drosophila melanogaster*) | *tubulin-GAL4* | Bloomington | 5138 | *y[1] w[*]; P{w[+mC]= tubP-GAL4}LL7/TM3, Sb[1] Ser[1]* |
| Genetic reagent (*Drosophila melanogaster*) | *elav-GAL4* | Bloomington | 77894 | *y[1] w[*] P{w[+mC]= elav(FRT.ewg)GAL4.eFeG}1* |

*Continued on next page*

Continued

| Reagent type (species) or resource | Designation | Source or reference | Identifiers | Additional information |
|---|---|---|---|---|
| Genetic reagent (Drosophila melanogaster) | nrv-GAL4 | Bloomington | 6799 | w[*]; P{w[+mC]=nrv2-GAL4.S}8 |
| Genetic reagent (Drosophila melanogaster) | D42-GAL4 | Bloomington | 8816 | w[*]; P{w[+mW.hs]=GawB}D42 |
| Genetic reagent (Drosophila melanogaster) | n-syb-GAL4 | Bloomington | 51635 | y[1] w[*]; P{w[+m*]=nSyb-GAL4.S}3 |
| Genetic reagent (Drosophila melanogaster) | repo-GAL4 | Bloomington | 7415 | w[1118]; P{w[+m*]=GAL4} repo/TM3, Sb[1] |
| Genetic reagent (Drosophila melanogaster) | 69B-GAL4 | Bloomington | 1744 | w[*]; P{w[+mW.hs]=GawB}69B |
| Genetic reagent (Drosophila melanogaster) | Mhc-GAL4 | Bloomington | 38464 | ; w[*]; P{w[+mC]=Mhc-RFP.F3-580} |
| Genetic reagent (Drosophila melanogaster) | Mef2-GAL4 | Bloomington | 26882 | w[*]; Kr[If-1]/CyO, P{w[+mC]=GAL4-Mef2.R}2, P{w[+mC]=UAS-mCD8.mRFP} |
| Genetic reagent (Drosophila melanogaster) | c179-GAL4 | Bloomington | 6450 | w[*]; P{w[+mW.hs]=GawB}c179 |
| Genetic reagent (Drosophila melanogaster) | how$^{24B}$-GAL4 | Bloomington | 1767 | w[*]; P{w[+mW.hs]=GawB}how[24B] |
| Genetic reagent (Drosophila melanogaster) | GMR-GAL4 | Bloomington | 9146 | w[1118]; P{GMR-GAL4.w[-]}2/CyO |
| Genetic reagent (Drosophila melanogaster) | nub-GAL4 | Bloomington | 86108 | w[*]; P{w[nub.PK]=nub-GAL4.K}2 |
| Genetic reagent (Drosophila melanogaster) | 5053 GAL4 | Bloomington | 2702 | w[*]; P{w[+mW.hs]=GawB} tey[5053A]/TM6B, Tb[+] |
| Genetic reagent (Drosophila melanogaster) | srmd710-GAL4 | Bloomington | 26663 | w[*]; P{w[+mW.hs]=GawB} sr[md710]/TM6B, Tb[1] |
| Genetic reagent (Drosophila melanogaster) | UAS dCNBP$^{RNAi}$ | VDRC | GD16283 | CNBP long hairpin on chromosome X |
| Genetic reagent (Drosophila melanogaster) | UAS dCNBP$^{RNAi}$ | VDRC | GD16284 | CNBP long hairpin on chromosome 2 |
| Genetic reagent (Drosophila melanogaster) | 2XUAS dCNBP$^{RNAi}$ | Antonucci et al., 2014 | GD16283+ GD16284 | - |
| Genetic reagent (Drosophila melanogaster) | dCNBP $^k$ | Kyoto | 203535 | y[1] w[67c23]; P{w[+mC]=GSV6}GS11716 / SM1 |
| Genetic reagent (Drosophila melanogaster) | UAS dCNBP-HA RNAi resistant | This study | – | Injection stock #BL 8622 |
| Genetic reagent (Drosophila melanogaster) | UAS hCNBP-FLAG | This study | – | Injection stock #BL 8622 |
| Genetic reagent (Drosophila melanogaster) | UAS Odc1$^{RNAi}$ | VDRC | GD30039 | Odc1 long hairpin on chromosome 3 |
| Genetic reagent (Drosophila melanogaster) | UAS Odc1$^{RNAi}$ | VDRC | GD30038 | Odc1 long hairpin on chromosome 2 |
| Genetic reagent (Drosophila melanogaster) | UAS Odc2$^{RNAi}$ | VDRC | KK104597 | Odc2 long hairpin on chromosome 2 |
| Genetic reagent (Drosophila melanogaster) | UAS Odc1+Odc2$^{RNAi}$ | This study | GD30038+ KK104597 | Long hairpin for Odc2 on chromosome 2 and for Odc1 on chromosome 3 |
| Genetic reagent (Drosophila melanogaster) | dOdc1 mutant | Bloomington | 56103 | y[1] w[*]; Mi{y[+mDint2]=MIC}Odc1[MI10996] |
| Genetic reagent (Drosophila melanogaster) | UAS dOdc1 | Gupta et al., 2013 | - | - |

| Reagent type (species) or resource | Designation | Source or reference | Identifiers | Additional information |
|---|---|---|---|---|
| Antibody | anti-CNBP (goat) | Abcam | ab48027, RRID:AB870003 | WB 1:1000 |
| Antibody | anti-ODC (rabbit) | ENZO | BML-PW8880-0100 RRID:AB_2156495 | WB 1:500 |
| Antibody | anti-Actin goat | Santa Cruz | sc-1616, RRID:AB630836 | WB 1:3000 |
| Antibody | anti-GFP (mouse) | Santa Cruz | sc-9996, RRID:AB_627695 | WB 1:3000 |
| Antibody | anti-Vinculin (mouse) | Santa Cruz | sc-73614, RRID:AB_1131294 | WB 1:3000 |
| Antibody | anti-FLAG-HRP | Sigma | A8592, RRID:AB_439702 | WB 1:500 |
| Antibody | anti-CNBP (mouse) | Agrobio (this study) | – | WB 1:1000 |
| Antibody | anti-HA-HRP (mouse) | Santa Cruz | sc-7392, RRID:AB_627809 | WB 1:2000 |
| Antibody | anti-vibrator (rabbit) | *Giansanti et al., 2006* | - | WB 1:3000 |
| Sequence-based reagent | *T7 CNBP FW Drosophila melanogaster* | This study | dsRNA primer | TAATACGACTCACTATAG GGAG GTCCGGGCGGCGTTGG |
| Sequence-based reagent | *T7 CNBP RV Drosophila melanogaster* | This study | dsRNA primer | TAATACGACTCACTATAG GGAG ATGTGTCCGGTGCGG |
| Sequence-based reagent | *dOdc1 Fw Drosophila melanogaster* | This study | PCR primer | TGGCAGCGATGACGTAAAGTT |
| Sequence-based reagent | *dOdc1 Rv Drosophila melanogaster* | This study | PCR primer | TGGTTCGGCGATTATGTGAA |
| Sequence-based reagent | *GAPDH Fw Drosophila melanogaster* | This study | PCR primer | CCTGGCCAAGGTCATCAATG |
| Sequence-based reagent | *GAPDH Rv Drosophila melanogaster* | This study | PCR primer | ATGACCTTGCCCACAGCCTT |
| Sequence-based reagent | dOdc1-IRES FW | This study | PCR primer | TAAGAATTCCTCGGAAAGATCTCAAC |
| Sequence-based reagent | dOdc1-IRES RW | This study | PCR primer | TTAGAATTCACAAGTCGT TGACTGATAAC |
| Chemical compound, drug | DFMO | Sigma | #D193 | |
| Commercial assay or kit | RevertAid H Minus First Strand cDNA Synthesis kit | Thermo Fisher Scientific | K1632 | |
| Chemical compound, drug | Putrescine | Sigma | #51799 | |
| Chemical compound, drug | Spermidine | Sigma | #S2626 | |
| Plasmids | plko SCR, plkoSh_cnbp | *D'Amico et al., 2015* | | |
| Plasmids | GFP | *Coni et al., 2020* | | |
| Plasmids | hODC-LUC | *D'Amico et al., 2015* | | |
| Plasmids | dOdc1-LUC | This study | | |
| Plasmids | HA-dCNBP | *Antonucci et al., 2014* | | |
| Plasmids | FLAG-hCNBP | *D'Amico et al., 2015* | | |
| Cell line | S2 | DGRC | Cat# 181, RRID:CVCL_Z992 | |
| Cell line | HEK-293T | ATCC | CRL-3216, RRID:CVCL_0063 | |

### *Drosophila* strains and rearing conditions

*Drosophila* stocks were maintained on standard fly food (25 g/L corn flour, 5 g/L lyophilized agar, 50 g/L sugar, 50 g/L fresh yeast, 2,5 mL/L Tegosept [10% in ethanol], and 2.5 mL/L propionic acid) at 25°C in a 12 hr light/dark cycle. All experiments were performed in the same standard conditions, at the temperature reported in figure legends.

The *2XUAS-dCNBP*$^{RNAi}$ strain used for *dCNBP* downregulation was already described in Antonucci et al., 2014. Essentially, are transgenic flies carrying two different *UAS-dCNBP*$^{RNAi}$ constructs (VDRC, ID 16283 and 16284) one on the X and one on the second chromosome, respectively.

The *UAS-Odc1*$^{RNAi}$ and the *UAS-Odc2*$^{RNAi}$ strains were also obtained from VDRC (ID 30039 and 104597) and similarly were combined to generate the strain *UAS-dOdc1-2*$^{RNAi}$ bearing both constructs to downregulate both isoform at the same time. *dCNBP*$^{k}$ is one of the P element insertions in the *CG3800* locus obtained from the Kyoto DGRC (#203535). The RNAi-resistant *dCNBP* gene carries synonymous substitutions in each residue of the region recognized by *UAS-dCNBP*$^{RNAi}$ and was synthesized by Genewiz (Sigma-Aldrich). The plasmids for inducible expression of RNAi-resistant *dCNBP-3HA* (abbreviated with *dCNBP-3HA-res*) were generated by cloning the 3 HA epitope CDS fused in-frame with the 3′ end of the *RNAi*-resistant *dCNBP* CDS into the UAS-attB vector (Genewiz, Sigma-Aldrich). The plasmids for inducible expression of the *hCNBP* counterpart were generated by cloning the FLAG epitope CDS fused in-frame with the 3′ end of the *hCNBP* CDS (CNBP-201 splice variant, CCDS 3056.1) into the UAS-attB vector (Genewiz, Sigma-Aldrich). The *dCNBP-3HA-res* or *UAS-hCNBP-FLAG* were injected in y$^1$ w$^{67c23}$; P{CaryP}attP2 embryos (BDSC Stock#8622); germline transformation was performed by Bestgene Inc (Chino Hills, CA) using standard procedures. *UAS–Odc-1* (Gupta et al., 2013). All the driver lines used have been previously described and available from the Bloomington stock center.

Spermidine or putrescine was added to standard food to a final concentration of 1 mM. For experiments, parental flies mated on either normal or spd+ or put+ food, and their progeny was allowed to develop on the respective food. DFMO was added to normal food to a final concentration of 5 mM/day.

## Climbing assays

The locomotion activity was measured by the DAM system (TriKinetics Inc, Waltham, MA), which allows a measure of fly locomotion capabilities based on their negative geotactic response, as the number of climbing performances in 30 min; 10–15 ageing-synchronized male flies (2–3 days, 7 days, or 15 days of age) were gathered and placed in each monitor for each genotype for each experiment. Briefly, the DAM system (TriKinetics Inc) records activity from individual flies maintained in sealed tubes placed in activity monitors. An infrared beam directed through the midpoint of each tube measures an 'activity event' each time a fly crosses the beam. The number of climbing events was scored for 30 min, tapping flies to the bottom every 40 s. Events detected over the course of each consecutive sampling interval are summed and recorded over the course of 30 min for each fly.

### *Drosophila* larval locomotion analyses

Larval locomotor activity was measured by counting the number of peristaltic contractions of third instar larvae performed within 1 min on the surface of a 1% agarose gel in a Petri dish; measurements were repeated five times for each larva, at least 10 larvae per genotype in each experiment.

## Immunoblot and antibodies

Protein extracts were derived from five third instar larvae, or cultured *Drosophila* S2 or human 293 cells, lysed in sample buffer, fractionated by SDS-PAGE and transferred to nitrocellulose membrane. Primary antibodies were: anti-CNBP goat (1:500; Abcam, Ab 48027); anti-Actin goat (1:1000; Santa Cruz, sc-1616); anti-ODC rabbit (1:500; Enzo Life Science, BML-PW8880-0100); anti-CNBP mouse (1:1000; generated by Agrobio for this work), anti-HA HRP (1:500; Santa Cruz, sc-7392), anti-GFP mouse (1:500; Santa Cruz, sc-9996); anti-FLAG HRP (1:1000; Sigma, A8592); anti-Vinculin mouse (1:1000; Santa Cruz, sc-73614); anti-Tubulin mouse (1:7000; Sigma, T-5168). As a secondary antibody, we used the appropriate HRP-conjugated antibody (GE Healthcare) diluted 1:5000 in 5% milk/PBS-Tween 0.1% (GE Healthcare). Detection was performed by using WesternBright ECL (K-12045-

D50, Advansta). Bands densitometric analysis was performed using the ImageJ software (version 1.50i). For DM2 patient biopsies, samples were lysed in SDS urea (50 mM Tris-HCl pH 7.8; 2% SDS, 10% glycerol, 1 mM EDTA, 6 M urea, 50 mM NaF, 5 mM $Na_2P_2O_7$) sonicated for 10 s, quantified by using a nanodrop and loaded on polyacrylamide gel. The study was carried out in line with the principles of the Declaration of Helsinki, and ethical approval was obtained from the Ethics Committee of the Fondazione Policlinico Universitario A Gemelli IRCCS Rome, Italy. Muscle biopsies used for this study were performed primarily for diagnostic purposes, after receiving an informed consent and consent to publish from all patients.

## RNA interference in S2 cell lines

S2 cells (DGRC, RRID:CVCL_Z232; tested negative for mycoplasma) were cultured at 25°C in Schneider's insect medium (Sigma) supplemented with 10% heat-inactivated fetal bovine serum (FBS, Gibco). RNAi treatments were carried out according to *Somma et al., 2008*. dsRNA-treated cells were grown for 4–5 days at 25°C, and then processed for biochemical analyses. PCR products and dsRNAs were synthesized as described in *Somma et al., 2008*. The primers used in the PCR reactions were 35 nt long and all contained a 5' T7 RNA polymerase binding site (5'-TAATACGAC TCACTATAGGGAGG-3') joined to a gene-specific sequence.

## RNA immunoprecipitation

S2 cells were plated in 75 cm$^2$ flask culture dishes and 72 hr later cells were crosslinked with 1% formaldehyde solution. Pellets were lysed with FA buffer (50 mM HEPES pH 7.5, 140 mM NaCl, 1 mM EDTA, 1% Triton X-100, 0.1% sodium deoxycholate, protease inhibitors, and 50 U/mL RNase inhibitor SupeRNase, #AM2694 Thermo Fisher Scientific) and sonicated.

For in vivo analysis, approximately 50 larval carcasses were UV-crosslinked ($3 \times 2000$ μJ/cm$^2$), homogenized on ice in 1 mL RCB buffer (50 mM HEPES pH 7.4, 200 mM NaCl, 2.5 mM MgCl$_2$, 0.1% Triton X-100, 250 mM sucrose, 1 mM DTT, 1× EDTA-free Complete Protease Inhibitors, 1 mM PMSF), supplemented with 300 U RNAseOUT (Invitrogen), and placed on ice for 30 min. The homogenate was sonicated on ice, at 80% power, five times in 20 s bursts with a 60 s rest in between using the Hielscher Ultrasonic Processor UP100H (100 W, 30 kHz) and centrifuged ($16,000 \times g$ for 5 min at 4°C).

Immunoprecipitation was performed by incubating the samples with anti-CNBP antibody or IgG overnight. Then, the samples were washed with RCB buffer four times, or with three different solutions for S2 extracts: low salt solution: 0.1% SDS, 1% Triton X-100 2 mM, EDTA 20 mM Tris-HCl pH 8, 150 mM NaCl, and 0.005 U/mL SuperRNAse (Thermo Fisher Scientific); high salt solution: 0.1% SDS, 1% Triton X-100, 2 mM EDTA, 20 mM Tris-HCl pH 8, 500 mM NaCl, and 0.005 U/mL Super-RNAse; LiCl buffer solution: 0.25 M LiCl, 1% NP40, 1% sodium deoxycholate, 1 mM EDTA, 10 mM Tris-HCl pH 8, and 0.005 U/mL SuperRNAse; TE wash buffer solution: 10 mM Tris-HCl pH 8, 1 mM EDTA, and 0.005 U/mL SuperRNAse, and then eluted with H$_2$O or elution buffer solution for S2 extracts: 1% SDS, 0.1 M NaHCO$_3$, SuperRNAse 50 U/mL. RNA was purified using Trizol reagent (15596026, Thermo Fisher Scientific), it was reverse-transcribed and dOdc was amplified by qPCR. Results were normalized on RPL11.

## RNA extraction and quantitative PCR

Total mRNA was isolated from S2 cells or *Drosophila* larvae by using Trizol reagent (15596026, Thermo Fisher Scientific) according to the manufacturer's instructions. RNA was reverse-transcribed (1 μg each experimental point) by using SensiFAST cDNA Synthesis Kit (BIO-65053, Bioline) and qPCR was performed as described (*Di Magno et al., 2020*) using SensiFast Sybr Lo-Rox Mix (BIO-94020, Bioline). The run was performed by using the Applied Biosystems (Waltham, MA) ViiA 7 Real-Time PCR System 36 instrument.

The following primers were used:

dOdc1 Fw: TGGCAGCGATGACGTAAAGTT;
dOdc1 Rv: TGGTTCGGCGATTATGTGAA;
dRPL11 Fw; CCATCGGTATCTATGGTCTGGA;
dRPL 11 Rv; CATCGTATTTCTGCTGGAACCA;
GFP Fw: GCAAAGACCCCAACGAGAAG;

GFP Rv: TTCTGATAGGCAGCCTGCAC;
dGADPH Fw: CCTGGCCAAGGTCATCAATG;
dGADPH Rv: ATGACCTTGCCCACAGCCTT;

## Polysome analysis

Polysomal fractionation from S2 cells was performed as described previously (*Coni et al., 2020*); S2 cells (interfered or not) were incubated 5 min with 100 µg/mL CHX, then washed with PBS and lysed with TNM buffer (10 mM Tris-HCl pH 7.4 or 7.5, 10 mM NaCl, 10 mM MgCl2, 1% Triton X-100), supplemented with 10 mM dithiothreitol, 100 µg/mL CHX, 1× PIC (1187358001 complete, EDTA free, Roche), and RiboLock RNase inhibitor (EO0382, Thermo Fisher Scientific). Lysates were incubated on ice for 10 min and then centrifuged at 2000 rpm for 5 min. Supernatants were loaded onto 15–50% sucrose gradients and centrifuged for 120 min in a Beckman SW41 rotor at 37,000 rpm at 4°C. Fractions were automatically collected, using Biorad-BioLogic LP/2110 (Hercules) monitoring the optical density at 260 nm. RNA was extracted from each fraction by using Trizol Reagent and dOdc mRNA was amplified by RT-qPCR. GAPDH mRNA was used for normalization.

## 293T lentiviral transduction and transfection

Lentivirus production was performed as described in *D'Amico et al., 2015*. Then human 293T cells were transduced with lentiviral particles of plkoSCR (Mission plko.1 puro; SHC002) or shCNBP human (Mission plko.1 puro TRCN0000311158, Sigma-Aldrich) at an MOI=5 for 72 hr. Then 293T cells SCR and shCNBP were transfected with plasmids encoding for dOdc and GFP for extra 24 hr, by using Dreamfect reagent according to manufacturer (DF41000 OZ, Biosciences). Cell extracts were analyzed through western blot and qPCR as indicated.

## Polyamine analysis

Polyamine content was determined by gas chromatography-mass spectrometry (GC-MS) and the values were normalized by the protein concentration. A pool of 10 third instar larvae for each genotype were resuspended in 0.2 M HClO and homogenized in an ice-bath using an ultra-turrax T8 blender. The homogenized tissue was centrifuged at 13,000× *g* for 15 min at 4°C; 0.5 mL of supernatant was spiked with internal standard 1,6-diaminohexane and adjusted to pH≥12 with 0.5 mL of 5 M NaOH. The samples were then subjected to sequential *N*-ethoxycarbonylation and *N*-pentafluoropropionylation. For DM2 samples, biopsies were also resuspended in 0.2 M HClO$_4$ and processed as described above. GC-MS analyses were performed with an Agilent 6850A gas chromatograph coupled to a 5973N quadrupole mass selective detector (Agilent Technologies, Palo Alto, CA). Chromatographic separations were carried out with an Agilent HP-5ms fused-silica capillary column. Mass spectrometric analysis was performed simultaneously in TIC (mass range scan from *m/z* 50 to 800 at a rate of 0.42 scans s–1) and SIM mode (put, *m/z* 405; spd, *m/z* 580, N1-acetyl-spm, *m/z* 637; spm, *m/z* 709).

## Immunostaining and confocal imaging

Larvae were dissected in ice-cold Ca$^{2+}$-free HL3 saline and fixed in 4% formaldehyde for 10 min and washed in PBS containing 0.05% Triton X-100 (PBST) for 30 min. After washing, larval fillets were stained with phalloidin-TRITC (1:300 diluted in PBST, Sigma) for 40 min at room temperature and subsequently washed for 3× 20 min with 0.05% PBST. Larvae were mounted in Vectashield containing DAPI (Vector Laboratories).

Confocal microscopy was performed with a Leica SP8 confocal microscope (Leica Microsystems, Germany). Confocal imaging of larval fillets was done using a z step of 0.5 µm. The following objective was used: 63× 1.4 NA oil immersion for confocal imaging. All confocal images were acquired using the LCS AF software (Leica, Germany). Images from fixed samples were taken from third instar larval fillets (segment A2, muscle 6/7).

## Luciferase assays

For luciferase assays, 293T cells were seeded and transfected for 24 hr by using Dreamfect Reagent (DF45000, OZ Bioscience) with the dOdc or human ODC (hODC) luciferase-renilla bicistronic reporters, HA-dCNBP or FLAG-hCNBP expression vectors, or pcDNA3 as an empty vector. Luciferase

reporter assay was performed by using the Firefly and Renilla Luciferase Single Tube Assay Kit (#30081–1, Biotium). Relative luciferase activity is expressed as the ratio of luciferase and renilla activity. The cloning of dOdc1 IRES was performed by PCR amplification of DNA from third instar larvae, by using the following oligos FW: TAAGAATTCCTCGGAAAGATCTCAAC, RW: TTAGAA TTCACAAGTCGT TGACTGATAAC. The amplicon was cloned in the backbone plasmid prl-Sammons (*D'Amico et al., 2015*). The cloning product was checked on agarose gel after enzymatic restriction with the EcoR1 enzyme and the final plasmid was verified by sequencing.

## Statistical analyses

Statistical analysis was performed using Prism six software (MacKiev). The Shapiro-Wilk test was used to assess the normal distribution of every group of different genotypes. Statistical differences for multiple comparisons were analyzed with the Kruskal-Wallis for non-parametric values or with one-way ANOVA for parametric values. The Dunn's or the Tukey's test was performed, respectively, as post hoc test to determine the significance between every single group. The Mann-Whitney U-test or the t-test were used for two groups' comparison of non-parametric or parametric values, respectively. A $p < 0.01$ was considered significant.

## Acknowledgements

This work was supported by AFM Telethon (French Muscular Dystrophy Association, project 21025). AIRC (Associazione Italiana Ricerca Cancro IG 17575), Sapienza University grant RM1181642798C54A and Italian Ministry of Education, Universities and Research, Dipartimenti di Eccellenza-L (232/2016), Istituto Pasteur, Fondazione Cenci-Bolognetti, Rome Italy. We thank Vittorio Padovano for his contribution in generating the mouse anti-dCNBP antibody. We thank Gianluca Cestra for critical reading of the manuscript.

## Additional information

### Funding

| Funder | Grant reference number | Author |
| --- | --- | --- |
| AFM-Téléthon | 21025 | Laura Ciapponi<br>Gianluca Canettieri |
| Associazione Italiana per la Ricerca sul Cancro | IG 17575 | Gianluca Canettieri |
| Sapienza Università di Roma | RM1181642798C54A | Gianluca Canettieri |
| Istituto Pasteur-Fondazione Cenci Bolognetti | Anna Tramontano | Gianluca Canettieri |

The funders had no role in study design, data collection and interpretation, or the decision to submit the work for publication.

### Author contributions

Sonia Coni, Conceptualization, Data curation, Formal analysis, Supervision, Investigation, Methodology, Writing - review and editing; Federica A Falconio, Data curation, Formal analysis, Investigation; Marta Marzullo, Data curation, Formal analysis, Supervision, Investigation, Methodology, Writing - review and editing; Marzia Munafò, Validation, Investigation; Benedetta Zuliani, Alessia Perna, Federica Mosti, Rosa Bordone, Alberto Macone, Tanja Matkovic, Investigation; Alessandro Fatica, Enzo Agostinelli, Supervision, Investigation, Methodology; Zaira Ianniello, Gabriella Silvestri, Investigation, Methodology; Stephan Sigrist, Supervision, Investigation, Methodology, Writing - review and editing; Gianluca Canettieri, Conceptualization, Supervision, Funding acquisition, Investigation, Writing - original draft, Writing - review and editing; Laura Ciapponi, Conceptualization, Data curation, Formal analysis, Supervision, Funding acquisition, Methodology, Writing - original draft, Project administration, Writing - review and editing

Author ORCIDs

Sonia Coni ⓘ https://orcid.org/0000-0002-0295-8904
Marta Marzullo ⓘ https://orcid.org/0000-0001-7229-1693
Marzia Munafò ⓘ https://orcid.org/0000-0002-2689-8432
Stephan Sigrist ⓘ https://orcid.org/0000-0002-1698-5815
Gianluca Canettieri ⓘ https://orcid.org/0000-0001-6694-2613
Laura Ciapponi ⓘ https://orcid.org/0000-0002-0817-1862

## Ethics

Human subjects: The study was carried out in line with the principles of the Declaration of Helsinki, and ethical approval was obtained from the Ethics Committee of the Fondazione Policlinico Universitario A. Gemelli IRCCS Rome, Italy. Muscle biopsies used for this study were performed primarily for diagnostic purposes, after receiving an informed consent and consent to publish from all patients.

## Decision letter and Author response

Decision letter https://doi.org/10.7554/eLife.69269.sa1
Author response https://doi.org/10.7554/eLife.69269.sa2

# Additional files

## Supplementary files

• Source data 1. Original Blots for all Western Blot experiments.

• Source data 2. Single Original Blots for all Western Blot experiments in tiff format.

• Source data 3. Western Blot quantifications as shown in *Figure 1E*, *Figure 1—figure supplement 1*, *Figure 2B*, *Figure 3A*, *Figure 3—figure supplement 1A*, *Figure 5A*, *Figure 5—figure supplement 1*, *Figure 6—figure supplement 3*, *Figure 1—figure supplement 4A*, *Figure 7—figure supplement 2*.

• Transparent reporting form

## Data availability

All data generated or analysed during this study are included in the manuscript and supporting files. Source data files have been provided for all figures.

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
