## [Decision Letter]

**Acceptance summary:**

This interesting study in *Drosophila* suggests that at least part of the pathology of DM2, a certain form of muscular dystrophy, is caused by defects in a gene that is required for the production of small molecules, called polyamines which are known to support muscle health and function. Observations in a *Drosophila* model of DM2 show that feeding with polyamines can restore muscle function, Taken together with observations muscle biopsies from human DM2 patients also show decreased levels of polyamines and ODC, a key enzyme in polyamine metabolism, this work raises possibility of using polyamines for therapy or prevention.

**Decision letter after peer review:**

Thank you for submitting your article "Translational control of polyamine metabolism by CNBP is required for locomotor function in *Drosophila melanogaster*" for consideration by *eLife*. Your article has been reviewed by 3 peer reviewers, including Mani Ramaswami as Reviewing Editor and Reviewer #1, and K VijayRaghavan as the Senior Editor.

Essential revisions:

1) One finding that is puzzling and could be better explained is shown in Figure 4 supplement: why is the reduction of putrescine levels the same when either the ODC1 or ODC2 gene is knocked down alone and when both genes are targeted at the same time? Shouldn't the double knock down show a stronger effect?

2) None of the western blots are quantified and there does not appear to be any replicates for most of them. This is important to show, especially when the authors comment on the levels of a particular protein being affected, it is hard to know how strong a reduction is without appropriate quantification relative to a housekeeping gene. Ideally, all western blots would need quantifying, but the most important ones would be Figure 1D, Figure 3A, Figure 3-S1, Figure 5A Figure 5S, Figure 6-S3.

3) Table 1, did the authors check the levels of knock-down for the neuronal drivers? RNAi in brain of flies is often difficult to achieve, it is important to determine this, to ensure that the reason there is no phenotype with neuronal drivers is not simply because there is no knock-down. Alternatively, the caveat should be acknowledged and it should be formally noted that potential brain function has not yet been excluded by the current analysis.

4) Lines 244-246, if patient tissue samples are available the authors should perform (and if not should indicate the need for) further direct analysis to formally demonstrate differential splicing in DM2 patients

5) Figure 7C there is no quantification of the CNBP knock-down when dOdc1 is over-expressed, this is important as the rescue could be due to a diluting out of the Gal4 driver (given there are now 3 transgenes in that fly), this would lead to a reduced knock-down and the rescue could be due to the CNBP levels increasing again.

6) For the ageing experiments in Figure 7 to be completely convincing, the authors would need to back-cross all their lines into a homogeneous back-ground 6 times, as back-ground variation has a very strong effect on speed of ageing. Also, to show that this reduction is not an effect of developmental defects, the authors should down-regulate CNBP post-eclosion, either using the Gal4/Gal80 system or the Gene-Switch system. Given how long these experiments would take, unless data are available, this section and relevant discussion should be duly abbreviated and/or qualified to acknowledge limitations of conclusions based on these data..

7) Lines 351-353: The authors use mutant CNBP for part of their characterisation, this would be a full animal knock-down, does this animal display muscle defects?

8) Gene names and abbreviations should be defined in the abstract and the first time they are mentioned in the text

9) Full genotypes should be written out in the figure legends for clarity

10) the conditions of fly rearing should be outlines in the methods (temperature/humidity/light dark cycle/food recipe) for stocks and experiments

11) Figure 1A represents a video, but this does not come across in a PDF, which is how most people will visualise the paper, maybe a plot/trace of the locomotor activity can be included and the videos included in supplemental methods.

---

## [Author Response]

Essential revisions:1) One finding that is puzzling and could be better explained is shown in Figure 4 supplement: why is the reduction of putrescine levels the same when either the ODC1 or ODC2 gene is knocked down alone and when both genes are targeted at the same time? Shouldn't the double knock down show a stronger effect?

We thank the referees and editors for the question raised. We agree that if both Odc1 and Odc2 cause a decrease of putrescine content, it would be expected that depletion of both isoforms should result in a more pronounced depletion of this polyamine. However, it should be pointed out that intracellular putrescine pool is not only the result of ODC-mediated ornithine decarboxylation but it is also produced from coupled acetylation/oxidation of spermine and spermidine and from the intake of extracellular polyamines, through specific transmembrane transporters (Casero et al., 2018). Hence, in case of complete blockade of ODC activity, putrescine could not be fully depleted thanks to these additional mechanisms. As for individual Odc1 and Odc2 knockdown, it is likely that both isoforms contribute to the intracellular putrescine pool and that stable ablation of each of them causes a comparable putrescine depletion as a result of activation of the compensatory mechanisms described above.

These issues have been discussed in the revised manuscript (lines 214-219).

2) None of the western blots are quantified and there does not appear to be any replicates for most of them. This is important to show, especially when the authors comment on the levels of a particular protein being affected, it is hard to know how strong a reduction is without appropriate quantification relative to a housekeeping gene. Ideally, all western blots would need quantifying, but the most important ones would be Figure 1D, Figure 3A, Figure 3-S1, Figure 5A Figure 5S, Figure 6-S3.

We agree that quantification of western blots band intensity is important. Thus, we have quantified all western blots using the ImageJ software. The graph representations have been included for each corresponding figure.

In particular: WB of Figure 1D has been moved to Figure 1E, which now includes the corresponding quantification graph; WB of Figure 2 has been moved to Figure 2B, which now includes the corresponding quantification graph; in Figure 3A, 3-S1A, 5A and 5-S1 the corresponding quantification graph of each WB has been included; the band quantifications of the WBs shown in both Figure 6B and D are shown in the new Figure 6-S3A and B; in Figure 6-S4 (before Figure 6-S3) the corresponding quantification graph of the WB has been included (Figure 6-S4A).

All row data obtained from quantifications and used to generate the column graphs have been included as source data.

3) Table 1, did the authors check the levels of knock-down for the neuronal drivers? RNAi in brain of flies is often difficult to achieve, it is important to determine this, to ensure that the reason there is no phenotype with neuronal drivers is not simply because there is no knock-down. Alternatively, the caveat should be acknowledged and it should be formally noted that potential brain function has not yet been excluded by the current analysis.

To verify that the absence of phenotype observed with the neuronal drivers is not due to a reduced efficacy of the dCNBP knockdown, we analyzed the expression levels of the dCNBP protein in both larval or adult brains when the UAS-CNBP RNAi constructs were expressed under the control of either the *n-syb* or *elav* GAL4 driver. These results are presented in an additional supplementary figure (Figure 1, figure supplement 1), where an immunoblot shows that both larval or adult brains are almost completely depleted of the dCNBP protein (bands quantification has also been included in the figure). These results have been also discussed in the revised manuscript (lines 148-149). Unfortunately, we were not able to test the depletion of dCNBP with other GAL4 drivers causing absence of phenotype, such as 5053-, sr^md710^- or repo- GAL4, because in those drivers the GAL4 expression is limited to few specific cells, and the presence of not-depleted cells could contaminate the immunoblot analysis.

4) Lines 244-246, if patient tissue samples are available the authors should perform (and if not should indicate the need for) further direct analysis to formally demonstrate differential splicing in DM2 patients

We agree that further experimentation to demonstrate the mechanism underlying the reduced levels of CNBP would be of interest. Unfortunately, the amount of samples (muscle biopsies from patients) was very little and we had to use all the available material to measure polyamines and protein content, as shown in the manuscript. Our analysis shows that in transgenic flies expressing quadruplet expansion the levels of CNBP were not reduced, indicating that the downregulation is not linked to RNA toxicity and suggesting that the reduced content of CNBP protein is likely the consequence of other mechanisms, specifically related to the intronic mutations occurring within the human *CNBP/ZNF9* gene.

Further studies with muscle samples from patients will be required to elucidate this issue.

These considerations have been added to the revised manuscript (lines 235-239).

5) Figure 7C there is no quantification of the CNBP knock-down when dOdc1 is over-expressed, this is important as the rescue could be due to a diluting out of the Gal4 driver (given there are now 3 transgenes in that fly), this would lead to a reduced knock-down and the rescue could be due to the CNBP levels increasing again.

We agree with referees that dilution of GAL4 by the presence of additional transgenes is a tangible problem in the UAS-GAL4 binary system. In order to address this point, we have quantified the expression levels of dCNBP protein in dCNBP interfered individuals, either in the presence or in the absence of the *UAS-Odc1* transgene. This result is included in an additional supplementary figure (named Figure 7—figure supplement 2), where an immunoblot shows that the downregulation of CNBP protein induced by the *UAS-RNAi* constructs is not affected by presence of the *UAS-dOdc1* construct (bands quantification has also been included in the figure). This result is discussed in the revised manuscript (lines 291-294).

6) For the ageing experiments in Figure 7 to be completely convincing, the authors would need to back-cross all their lines into a homogeneous back-ground 6 times, as back-ground variation has a very strong effect on speed of ageing. Also, to show that this reduction is not an effect of developmental defects, the authors should down-regulate CNBP post-eclosion, either using the Gal4/Gal80 system or the Gene-Switch system. Given how long these experiments would take, unless data are available, this section and relevant discussion should be duly abbreviated and/or qualified to acknowledge limitations of conclusions based on these data..

We agree with referees that backcrossing our lines would exclude the influence of the genetic background heterogeneity on aging and that CNBP silencing post-eclosion would exclude the effects of developmental defects in aging acceleration. However, as noted by referees these experiments would take too much time to get clear and exhaustive answers. As suggested by the referees, we have acknowledged the limitation of these experimental conditions in the revised manuscript. We have significantly toned down our statements and highlighted that this is an interesting observation that poses attractive hypotheses (lines 305/308-312).

7) Lines 351-353: The authors use mutant CNBP for part of their characterisation, this would be a full animal knock-down, does this animal display muscle defects?

As suggested, we have performed analysis of muscle structure in mutant CNBP flies. As shown in the additional panel included in Figure 7—figure supplement 4 (before Figure 7—figure supplement 3) we did not see clear differences in muscle morphology in CNBP mutant larvae compared to controls. In this regard, it must be pointed out that while CNBP mutant animals die early during larval development (2nd instar), a significant fraction of CNBP homozygous KO mice are viable at birth, when the muscle phenotype is detected (Wei et al., 2018). Hence, it is possible that such a short survival of CNBP mutant flies does not allow sufficient time for the muscle alterations to fully develop and be appreciated. Moreover, we cannot exclude that an early requirement of CNBP for muscle development might be overshadowed by the presence of maternal contribution provided by the heterozygous mother flies. Further studies will help to understand this issue.

These points have been included in the revised manuscript (lines 351-360).

8) Gene names and abbreviations should be defined in the abstract and the first time they are mentioned in the text9) Full genotypes should be written out in the figure legends for clarity10) the conditions of fly rearing should be outlines in the methods (temperature/humidity/light dark cycle/food recipe) for stocks and experiments

We have defined all abbreviations in the abstract and the first time they are mentioned. In figure legends all genotypes have been fully written and Materials and methods now specify the conditions of fly rearing (lines 425-428). Moreover, specific experimental conditions are always reported in the figure legend.

11) Figure 1A represents a video, but this does not come across in a PDF, which is how most people will visualise the paper, maybe a plot/trace of the locomotor activity can be included and the videos included in supplemental methods.

We completely agree with referees that it is essential to represent the biological message displayed by the fly's videos as a still image, compatible with a PDF version of the manuscript. To this end, we took advantage of a specific ImageJ plugin, named animal tracker, that allowed us to analyze the recorded fly movement and to quantify both fly speed and distance covered in one minute. These data have been included in a new Figure 1A and discussed in the revised manuscript (lines 153-156).